# Generation of transgenic mice expressing a FRET biosensor, SMART, that responds to necroptosis

Shin Murai [1], Kanako Takakura[2], Kenta Sumiyama [3,12], Kenta Moriwaki [1], Kenta Terai [4], Sachiko Komazawa-Sakon [1], Takao Seki [1], Yoshifumi Yamaguchi [5], Tetuo Mikami[6], Kimi Araki [7,8], Masaki Ohmuraya[9], Michiyuki Matsuda[4,10,11] & Hiroyasu Nakano [1 ✉]

Necroptosis is a regulated form of cell death involved in various pathological conditions, including ischemic reperfusion injuries, virus infections, and drug-induced tissue injuries. However, it is not fully understood when and where necroptosis occurs in vivo. We previously generated a Forster resonance energy transfer (FRET) biosensor, termed SMART (the sensor for MLKL activation by RIPK3 based on FRET), which monitors conformational changes of MLKL along with progression of necroptosis in human and murine cell lines in vitro. Here, we generate transgenic (Tg) mice that express the SMART biosensor in various tissues. The FRET ratio is increased in necroptosis, but not apoptosis or pyroptosis, in primary cells. Moreover, the FRET signals are elevated in renal tubular cells of cisplatin-treated SMART Tg mice compared to untreated SMART Tg mice. Together, SMART Tg mice may provide a valuable tool for monitoring necroptosis in different types of cells in vitro and in vivo.

[1] Department of Biochemistry, Toho University School of Medicine, 5-21-16 Omori-Nishi, Ota-ku, Tokyo 143-540, Japan. [2] Imaging Platform for Spatio-Temporal Regulation, Graduate School of Medicine, Kyoto University, Kyoto 606-8501, Japan. [3] Laboratory for Mouse Genetic Engineering, RIKEN Center for Biosystems Dynamics Research, Suita 565-0874, Japan. [4] Research Center for Dynamic Living Systems, Graduate School of Biostudies, Kyoto University, Kyoto 606-8501, Japan. [5] Hibernation metabolism, physiology, and development Group, Environmental Biology Division, Institute of Lo600-601. Temperature Science, Hokkaido University, Kita 19, Nishi 8, Kita-ku, Sapporo, Hokkaido 060-0819, Japan. [6] Department of Pathology, Toho University School of Medicine, 5-21-16 Omori-Nishi, Ota-ku, Tokyo 143-8540, Japan. [7] Center for Animal Resources and Development, Kumamoto University, 2-2-1 Honjo, Chuo-ku, Kumamoto 860-0811, Japan. [8] Center for Metabolic Regulation of Healthy Aging, Kumamoto University, 1-1-1, Honjo, Chuo-ku, Kumamoto 860-8556, Japan. [9] Department of Genetics, Hyogo College of Medicine, Nishinomiya, Hyogo 663-8501, Japan. [10] Department of Pathology and Biology of Diseases, Graduate School of Medicine, Kyoto University, Kyoto 606-8501, Japan. [11] Institute for Integrated Cell-Material Sciences, Kyoto University, Kyoto 606-8501, Japan. [12]Present address: Laboratory of Animal Genetics and Breeding, Graduate School of Bioagricultural Sciences, Nagoya University, Chikusa, Nagoya 464-8601, Japan. ✉email: hiroyasu.nakano@med.toho-u.ac.jp

Necroptosis is a regulated form of cell death characterized by necrotic morphology. Necroptosis is involved in various pathological conditions, including tumor necrosis factor (TNF)-mediated systemic inflammatory response syndrome (SIRS), ischemic reperfusion injury, neurodegeneration, and drug-induced tissue injuries[1,2]. Necroptosis is induced by death ligands (e.g., TNF, Fas ligand, and TRAIL), polyinosinic-polycytidylic acid, lipopolysaccharide (LPS), interferon (IFN)α/β, IFNγ, some virus infections, and endogenous Z-form nucleic acids[3–5]. Among the various signaling pathways that lead to necroptosis, the TNF-mediated signaling pathway is one of the most extensively investigated[6,7]. In most cell types, TNF induces cell survival and inflammatory cytokine expression by activating the transcription factor, nuclear factor kappa B (NF-κB). However, under certain conditions, NF-κB activation is compromised, for example, when transforming growth factor β (TGFβ)-activated kinase 1 (TAK1) or inhibitor of apoptosis proteins (IAPs) are inhibited. Under those conditions, TNF induces apoptosis or necroptosis in a context-dependent manner[6,7]. Of note, under normal conditions, caspase 8 suppresses necroptosis by cleaving and inactivating receptor-interacting kinase 1 (RIPK1), an essential component of the TNF-induced necroptosis pathway[8,9]. When caspase 8 activity is blocked, either chemically, genetically, or by a viral infection, RIPK1 recruits RIPK3 via a homotypic interaction to form a necroptosis-inducing signaling complex, termed the necrosome[10–12].

In necrosomes, RIPK3 is activated by autophosphorylation, which leads to the formation of a higher-order amyloid-like RIPK1-RIPK3 hetero-oligomer[13,14]. Activated RIPK3 subsequently phosphorylates mixed lineage kinase domain-like pseudokinase (MLKL) that exists as a monomer in unstimulated cells[15,16]. MLKL is composed of an N-terminal four-helix bundle (4-HB) domain and a C-terminal pseudokinase (KL) domain[17,18]. While the N-terminal 4-HB domain binds to membrane phospholipids[15,19], the C-terminal KL domain is responsible for binding to RIPK3. Phosphorylated MLKL undergoes a large conformation change and disengages from RIPK3[20], resulting in the generation of the trimer (for mice MLKL) or tetramer (for human MLKL)[21–23]. Then, oligomerized MLKL translocates to the membrane and induces membrane rupture.

The ruptured membranes of dying cells release intracellular contents, which are referred to as damage-associated molecular patterns (DAMPs). DAMPs comprise various molecules, including heat shock proteins (HSPs), high mobility group protein B1 (HMGB1), ATP, and histones. Thus, DAMPs have pleiotropic functions that become active in a context-dependent manner[24,25]. However, it remains unclear when and where cells undergo necroptosis and release DAMPs. We previously developed a Forster resonance energy transfer (FRET) biosensor that could specifically monitor necroptosis, which we termed the sensor for MLKL-activation by RIPK3 based on FRET (SMART)[26]. To visualize the release of DAMPs at single-cell resolution, we modified a new technology termed Live-Cell Imaging for Secretion activity (LCI-S). This technology comprises a fluorescence-conjugated, antibody-based sandwich ELISA system and total inverted reflection fluorescence microscopy[27,28]. By combining the SMART and LCI-S technologies, we could visualize the execution of necroptosis and the release of HMGB1 (a nuclear DAMP) at single-cell resolution[26].

Here, we aimed to apply the SMART biosensor to in vivo experiments. We generated transgenic (Tg) mice that stably expressed SMART in various tissues. SMART Tg mice grew to adulthood without any apparent abnormality and were fertile. Consistent with previous results in tumor cell lines that expressed SMART, the FRET to cyan fluorescent protein (FRET/CFP) ratio was increased in peritoneal macrophages and murine embryonic fibroblasts (MEFs) from SMART Tg mice along with the progression of necroptosis, but not apoptosis or pyroptosis. Moreover, the increase in the FRET/CFP ratio was abrogated in peritoneal macrophages from SMART $Ripk3^{-/-}$ or SMART $Mlkl^{-/-}$ mice following stimuli that induced necroptosis in wild-type macrophages. These results further substantiated that RIPK3 and MLKL were indispensable for the increase in the FRET/CFP ratio of SMART along with progression of necroptosis in primary cells. Furthermore, in a cisplatin-induced kidney injury model, the averaged FRET/CFP ratios of renal proximal tubule cells were higher in cisplatin-treated SMART Tg mice than those in untreated SMART Tg mice. Collectively, we conclude that SMART Tg mice may be a valuable tool for monitoring conformational changes of MLKL along with the progression of necroptosis in vivo.

## Results

**Generation of SMART Tg mice.** The SMART is an intramolecular FRET biosensor. It comprises a fragment of the KL domain of MLKL containing α1 to α4 helices, placed between the enhanced cyan fluorescent protein (ECFP) sequence and the modified yellow fluorescent protein (Ypet) sequence[26,29] (Fig. 1a). The fluorescent proteins serve as the FRET donor and acceptor, respectively. In the presence of stimuli that induce necroptosis, activated RIPK3 binds and phosphorylates MLKL, resulting in large conformational changes and oligomerization of MLKL. Oligomers of MLKL elicit conformational changes in SMART, which increases the FRET efficiency, and the fluorescent signal changes from ECFP to Ypet (Fig. 1a).

To visualize necroptosis under various pathological conditions in vivo, we used the transposon system to generate transgenic mice that expressed SMART under the CAG promoter, as described previously[30]. SMART Tg mice grew without any apparent abnormality, and they were fertile. These mice stably expressed SMART in various tissues, including the liver, spleen, kidney, small intestine, and colon (Fig. 1b). To test whether SMART could respond to necroptosis in primary cells, as previously demonstrated in tumor cell lines[26], we isolated peritoneal macrophages from SMART Tg mice. Briefly, we performed an intraperitoneal injection of thioglycollate in mice to recruit inflammatory macrophages into the peritoneal cavity; then, 4 days later, we collected the macrophages. Approximately 80% of the collected peritoneal cells expressed both CD11b and F4/80; therefore, these cells were considered macrophages (Fig. 1c and Supplementary Fig. 1). In addition, the expression of yellow fluorescent protein (YFP) confirmed that these macrophages expressed SMART (Fig. 1c).

When macrophages are treated with the IAP inhibitor, BV6, macrophages produce TNF that can result in TNF-dependent apoptosis or necroptosis in a context-dependent manner[31,32]. In the presence of the caspase inhibitor, zVAD, BV6 treatment induces necroptosis. When we stimulated macrophages with BV6/zVAD, the release of lactate dehydrogenase (LDH), a hallmark of membrane rupture, gradually increased. However, this increase was completely abolished when cells were treated with BV6/zVAD in the presence of the RIPK3 inhibitor, GSK'872 (Fig. 1d). Live-cell imaging revealed that the FRET/CFP ratio was gradually increased in cells treated with BV6/zVAD, and then it abruptly declined (Fig. 1e, f and Supplementary Movie 1). The FRET/CFP ratio decline could have been due to the release of the SMART biosensor through the ruptured membrane. In contrast, the FRET/CFP was not increased in untreated cells or cells treated with BV6/zVAD/GSK'872 (Fig. 1e, f).

**SMART does not respond to pyroptosis or apoptosis.** We previously reported that SMART could sense conformational changes of MLKL along with the progression of necroptosis, but not apoptosis,

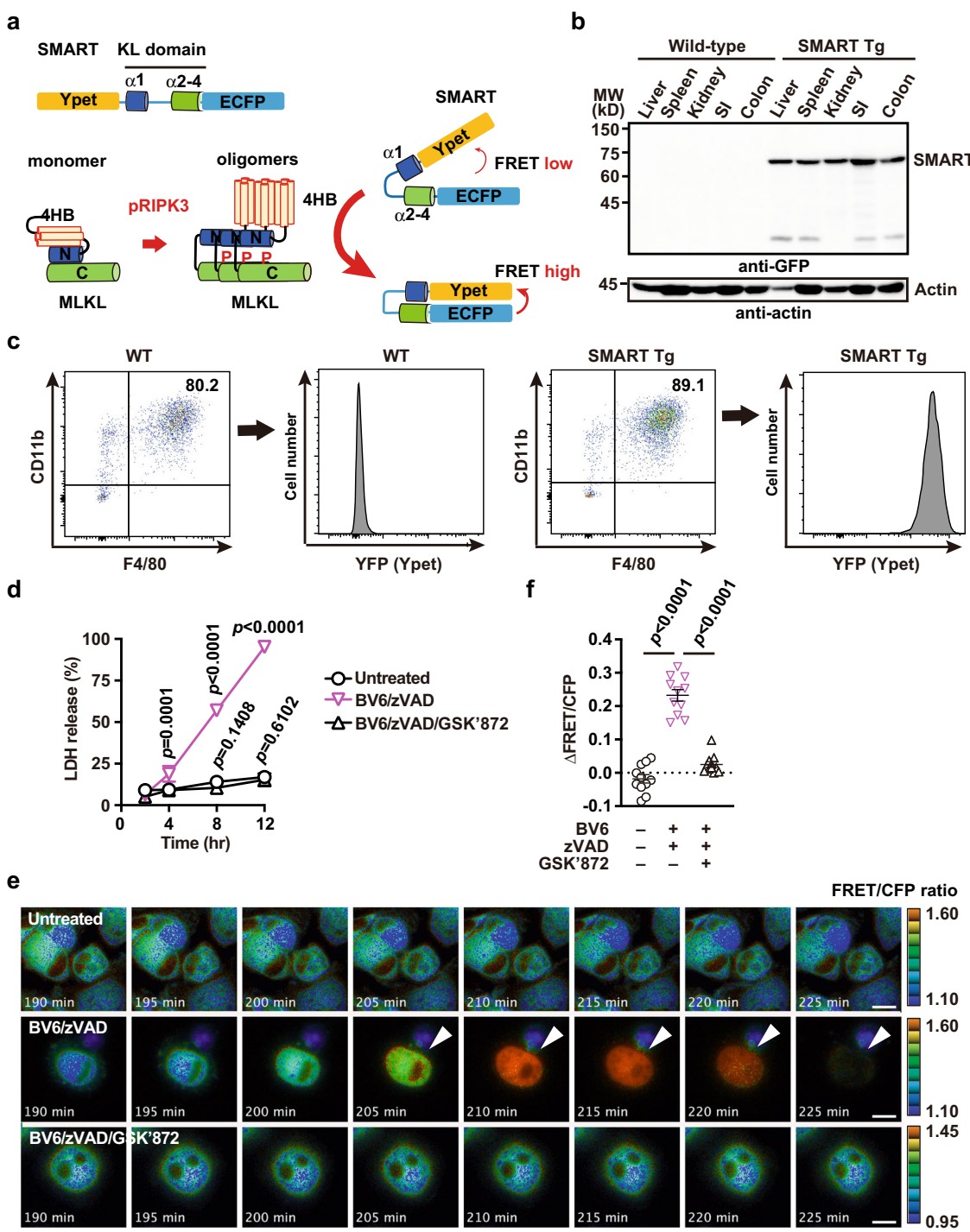

in various human and murine cell lines[26]. However, it remained unclear whether SMART responded to pyroptosis. To address this issue, we pretreated peritoneal macrophages from SMART Tg mice with LPS, followed by nigericin stimulation. LPS plus nigericin, but not LPS alone, caused a robust release of LDH from macrophages (Fig. 2a), which suggested that the macrophages died by pyroptosis. The FRET/CFP ratio was not increased; instead, it eventually decreased in cells treated with LPS/nigericin compared to cells treated with LPS alone (Fig. 2b, c and Supplementary Movie 2).

Priming macrophages with LPS and subsequently stimulating them with the TAK1 inhibitor, 5z-7-oxozeaneaeol (5z7Ox), can result in apoptosis, necroptosis, or pyroptosis, depending on the context[33–35]. To induce necroptosis selectively, we primed

macrophages with LPS, then stimulated them with 5z7Ox in the presence of zVAD. In parallel experiments, after priming, we stimulated macrophages with 5z7Ox in the presence of GSK'872 to induce apoptosis or pyroptosis. Macrophage viability decreased under both of the latter conditions, but viability was maintained when cells were stimulated with LPS/5z7Ox/GSK'872/zVAD, which blocked apoptosis, necroptosis, and pyroptosis (Fig. 2d). Of note, the FRET/CFP ratio increased in cells stimulated with LPS/5z7Ox/zVAD (necroptotic condition), but not in cells treated with LPS/5z7Ox/GSK'872 (apoptotic or pyroptotic conditions) (Fig. 2e, f). These results suggested that SMART could sense necroptosis, but not apoptosis or pyroptosis, in primary macrophages.

**Fig. 1 Generation of SMART Tg mice. a** Depiction of the SMART biosensor structure (*top*) and its activation mechanism (bottom). SMART is composed of N-terminal Ypet, a modified α1 to α4 helices of the KL domain of MLKL, and C-terminal ECFP. Upon necroptosis induction, activated and phosphorylated RIPK3 (pRIPK3) phosphorylates MLKL, resulting in oligomer formation of MLKL. Then, oligomers of MLKL induces conformational changes of SMART, possibly through the interaction, thereby increasing FRET efficiency. KL pseudokinase domain, Ypet modified yellow fluorescent protein, ECFP enhanced cyan fluorescent protein, pRIPK3 phosphorylated RIPK3. **b** Western blots probed with anti-GFP antibody show expression of the SMART biosensor in various murine tissues. Tissue extracts were prepared from the indicated organs of 8-week-old wild-type or SMART Tg mice. Results are representative of two independent experiments. **c** Mice were intraperitoneally injected with thioglycollate, then peritoneal cells were recovered by washing the peritoneal cavity with ice-cold PBS on day 4 after injection. Isolated cells were stained with the indicated antibodies and analyzed with flow cytometry. The percentages of CD11b+F4/80+ cells indicate the fraction of macrophages; the levels of YFP detected in these cell populations indicate the expression of SMART. Results are representative of three independent experiments. WT wild-type. **d** Peritoneal macrophages from SMART Tg mice were untreated or stimulated with BV6 (1 μM) + zVAD (20 μM) or BV6 (1 μM) + zVAD (20 μM) + GSK'872 (5 μM) for the indicated times. Cell death was assessed with the LDH release assay. Results are mean ± SD of triplicate samples, and they are representative of five independent experiments. **e, f** Peritoneal macrophages derived from SMART Tg mice were stimulated as described in **d**, and FRET/CFP ratios were calculated. Pseudocolored images show cellular changes in FRET/CFP ratio values in response to the indicated stimulations (**e**). FRET/CFP responses are color-coded according to the color scales (right). White arrowheads indicate cells undergoing necroptosis. Scale bars, 20 μm. Maximum changes detected in the FRET/CFP ratios (**f**). Results are mean ± SE ($n = 11$ cells per condition). Each dot indicates an individual cell. Results are representative of four independent experiments. Statistical significance was determined with two-way ANOVA with Dunnett's multiple comparison test (**d**) or one-way ANOVA with Turkey's multiple comparison test (**f**).

## Crucial role for RIPK3 and MLKL in the FRET response of SMART in macrophages following necroptosis induction.

Necroptosis depends on the activities of RIPK3 and MLKL[3,36]. We next tested whether the FRET/CFP ratio was increased in primary macrophages in a RIPK3- or MLKL-dependent manner, as observed in tumor cell lines[26]. To that end, we generated SMART Tg mice with $Ripk3^{-/-}$ or $Mlkl^{-/-}$ genetic backgrounds (SMART $Ripk3^{-/-}$ or SMART $Mlkl^{-/-}$ mice). We prepared peritoneal macrophages from SMART $Ripk3^{-/-}$ mice or SMART $Mlkl^{-/-}$ mice. In contrast to macrophages from SMART WT mice, BV6/zVAD-induced LDH release was abrogated in macrophages from SMART $Ripk3^{-/-}$ mice and SMART $Mlkl^{-/-}$ mice (Fig. 3a). Accordingly, the increase in the FRET/CFP ratio was abolished in macrophages from SMART $Ripk3^{-/-}$ mice and SMART $Mlkl^{-/-}$ mice after BV6/zVAD stimulation compared to those from SMART WT mice (Fig. 3b, c). These results showed that RIPK3 and MLKL are indispensable for the increase in the FRET/CFP ratio in primary macrophages following necroptosis induction.

## SMART responds to necroptosis, but not apoptosis, in primary MEFs.

To extend our observations in peritoneal macrophages to other cell types, we next stimulated primary MEFs derived from SMART Tg mice. We found that TNF/BV6/zVAD stimulation did not cause cell death in primary MEFs (pMEFs), but did cause cell death in immortalized MEFs (iMEFs) (Fig. 4a, b). We noticed that MLKL expression was much lower in pMEFs than in iMEFs (Fig. 4c). Because MLKL expression is induced by interferons[37,38], we treated pMEFs with IFNβ. MLKL expression was strongly enhanced; accordingly, IFNβ-pretreated pMEFs died in response to TNF/BV6/zVAD stimulation (Fig. 4d). In addition, the FRET/CFP ratio was increased in IFNβ-pretreated pMEFs after TNF/BV6/zVAD stimulation, but this increase was abolished in the presence of GSK'872 (Fig. 4e, f and Supplementary Movie 3). Moreover, when IFNβ-pretreated pMEFs were stimulated with TNF/BV6/GSK'872, apoptosis was induced, but the FRET/CFP ratio was not increased (Fig. 4g, h). This result suggested that SMART did not respond to apoptosis in pMEFs.

## Administration of cisplatin induces necroptosis in renal proximal tubular cells.

We tested whether in vivo imaging of necroptosis was feasible with SMART Tg mice. In contrast to apoptosis, which normally occurs in different developmental stages, necroptosis does not play a major role in the development[39,40]. However, necroptosis occurs in vivo in murine models of TNF-induced SIRS, ischemic reperfusion injuries, and cisplatin-induced kidney injuries[41–43]. In contrast to several previous studies, which showed that intestinal epithelial cells die by necroptosis after injecting high doses of TNF[41,42], we found that only small numbers of intestinal epithelial cells were pRIPK3-positive (Supplementary Fig. 2a). Moreover, it was difficult to measure the FRET/CFP ratio of SMART in an individual epithelial cell continuously with two-photon excitation microscopy due to constant peristalsis of the small intestine (Supplementary Fig. 2b, c and Supplementary Movie 4).

Therefore, we focused on cisplatin-induced kidney injury. Previous studies reported that administration of cisplatin induces renal proximal tubular cell necroptosis, and this injury is attenuated in $Ripk3^{-/-}$ mice[43,44]. We found that cisplatin injections resulted in gradual increases in blood urea nitrogen (BUN) and serum creatinine in wild-type (WT) mice (Fig. 5a). These signs were accompanied by kidney damage, such as the dilation of the proximal tubule lumens (Fig. 5b). We then investigated whether phospho-RIPK3 (pRIPK3)-positive cells, a hallmark of necroptosis, appeared in the kidney after cisplatin injection. We found that, after the cisplatin injection, the numbers of pRIPK3-positive cells gradually increased and peaked on day 2 in the renal cortex of wild-type mice (Fig. 5c, e). We also found that cleaved caspase 3 (CC3)-positive cells appeared in the kidneys of wild-type mice after the cisplatin injection (Fig. 5d, f). These findings suggested that cisplatin-induced both apoptosis and necroptosis in the kidney. In contrast to the previous studies[43,44], cisplatin-induced kidney injury was not attenuated, but rather exacerbated in $Ripk3^{-/-}$ mice compared to WT mice (Fig. 5g). Of note, pRIPK3+ cells were completely abolished in the kidneys of $Ripk3^{-/-}$ mice (Fig. 5h), but the numbers of CC3+ cells were increased in the kidneys of $Ripk3^{-/-}$ mice (Fig. 5i, j).

## Crucial role for RIPK3 in the FRET response of SMART in renal proximal tubular cells after cisplatin administration.

We then tested whether the FRET/CFP ratio was increased in SMART Tg mice with cisplatin-induced kidney injuries. Given that pRIPK3-positive cells peaked on day 2 after the cisplatin injection (Fig. 5c, e, h), we analyzed necroptosis by in vivo imaging on day 2. Almost all proximal tubule lumens were patent in untreated mice, and these proximal tubular cells did not show an increased FRET/CFP ratio (Fig. 6a, upper panel). However, in cisplatin-treated mice, many proximal tubule lumens were occluded, and several proximal tubules showed an increased FRET/CFP ratio on day 2 (Fig. 6a, lower panel). To investigate the kinetics of the occlusion of proximal tubule lumens and the

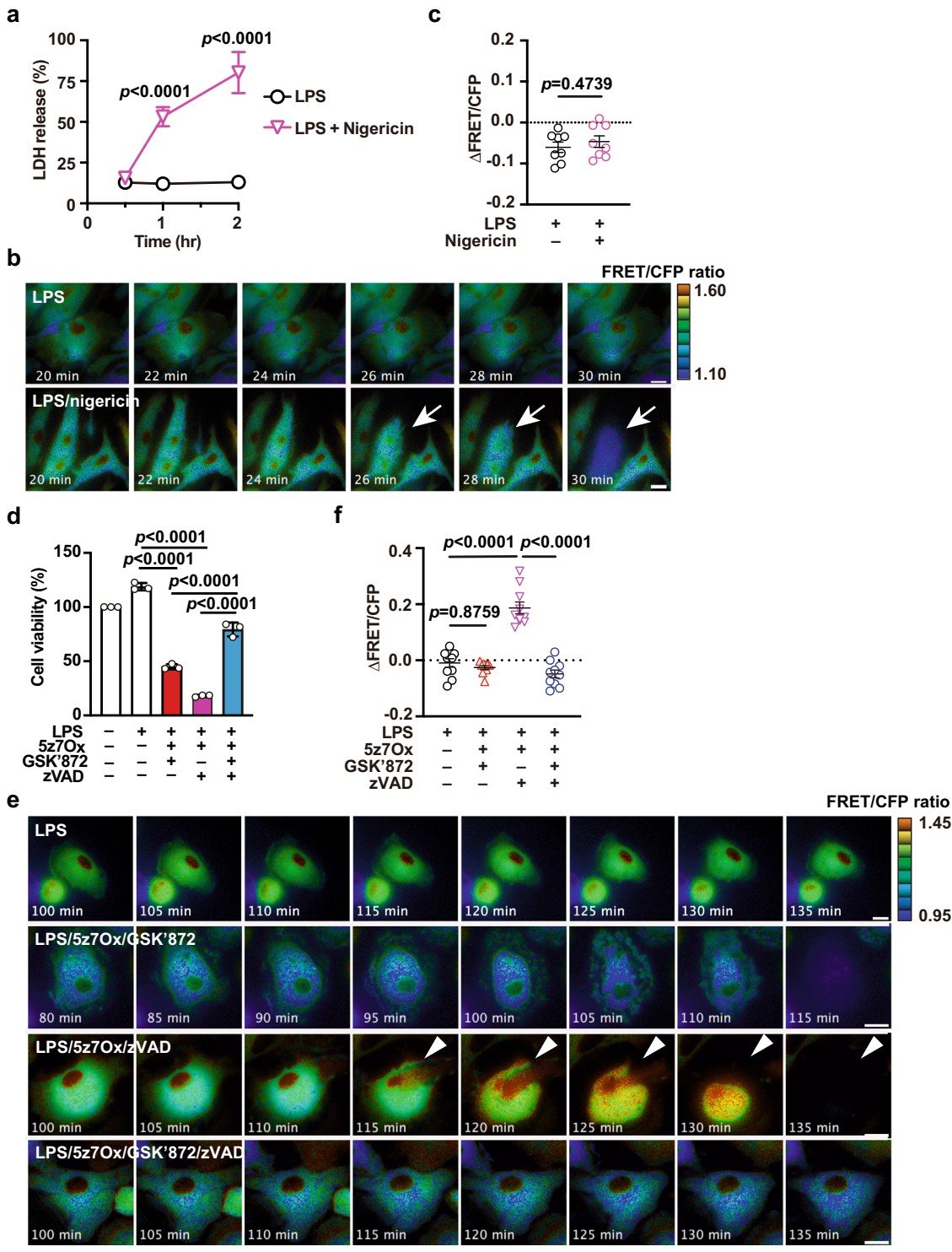

increased FRET/CFP ratio, we visualized these two events by live imaging following cisplatin injection on day 2. In untreated mice, most proximal tubule lumens remained patent during the imaging periods (Supplementary Movie 5). In sharp contrast, a few proximal tubule lumens were patent at the start of the imaging, but these tubule lumens were eventually occluded during the imaging period (Fig. 6b and Supplementary Movie 6). The FRET/CFP ratio of proximal tubular cells was increased and peaked at ~100 min after the occlusion of the proximal tubule lumens (Fig. 6b, c). These results suggested that the increase in the FRET/CFP ratio may be associated with the occlusion of the tubule

lumens and that SMART responded to necroptosis of the proximal tubular cells.

To evaluate the increase in the FRET/CFP ratio in the kidneys more quantitatively, we calculated the averaged FRET/CFP ratios of each tubule in the kidneys of untreated and cisplatin-treated mice, during the same period, on day 2 after the injection. We found significantly higher averaged FRET/CFP ratios in the kidneys of cisplatin-treated mice compared to the kidneys of untreated mice (Fig. 6d, e). In contrast, the averaged FRET/CFP ratios in SMART $Ripk3^{-/-}$ mice after the cisplatin injection were comparable to those observed in untreated SMART $Ripk3^{-/-}$

**Fig. 2 SMART does not respond to pyroptosis or apoptosis. a** Peritoneal macrophages isolated from SMART mice were pretreated with LPS (10 ng/mL) for 4 h, then stimulated with nigericin (10 μM) for the indicated times. Cell death was determined with the LDH release assay. Results are mean ± SD of triplicate samples and representative of three independent experiments. **b, c** Peritoneal macrophages were stimulated as described in **a**, and the FRET/CFP ratio was calculated. Pseudocolored images show cellular changes in FRET/CFP ratio values in response to the indicated stimulations (**b**). FRET/CFP responses are color-coded according to the color scales (right). White arrows indicate cells undergoing pyroptosis. Scale bars, 20 μm. Maximum changes detected in the FRET/CFP ratios (**c**). Results are mean ± SE (n = 8 cells per condition). Each dot indicates an individual cell. Results are representative of three independent experiments. **d** Peritoneal macrophages from SMART Tg mice were pretreated with LPS (10 ng/mL) for 4 h, then stimulated with the indicated combination of agents (125 nM 5z7Ox, 5 μM GSK'872, 20 μM zVAD) for 4 h. Cell viability was determined with the WST assay. Results are mean ± SD of triplicate samples. Representative results of four independent experiments. **e, f** Peritoneal macrophages were stimulated as described in **d**. Macrophages are displayed as pseudocolor images to show cellular changes in FRET/CFP ratio values in response to the indicated stimulations (**e**). FRET/CFP responses are color-coded according to the color scales (right). White arrowheads indicate cells undergoing necroptosis. Scale bars, 20 μm. Maximum changes detected in the FRET/CFP ratios of the cells (**f**). Results are mean ± SE (n = 10 cells per condition). Each dot indicates an individual cell. Results are representative of two independent experiments. Statistical significance was determined with two-way ANOVA with Sidak's multiple comparison test (**a**), the unpaired two-tailed Student t-test (**c**), or one-way ANOVA with Tukey's multiple comparison analysis (**d, f**).

mice (Fig. 6f, g). To our surprise, the averaged FRET/CFP ratios in renal tubular epithelial cells of cisplatin-treated SMART $Mlkl^{-/-}$ mice were higher than those of untreated SMART $Mlkl^{-/-}$ mice (Fig. 6h, i). This suggested that SMART may respond to conformational changes of RIPK3 in renal tubular epithelial cells of cisplatin-treated SMART $Mlkl^{-/-}$ mice. Consistently, pull-down assays in HEK293 cells expressing SMART along with MLKL or RIPK3 revealed a physical interaction of SMART with RIPK3 as well as MLKL (Supplementary Fig. 3). Together, SMART senses conformational changes of MLKL in primary macrophages and several tumor cell lines as previously reported[26], but also senses conformational changes of RIPK3 in renal tubular epithelial cells.

## Discussion

In the present study, we generated transgenic mice that constitutively expressed the FRET biosensor, SMART, in various tissues. Consistent with our previous study[26], we could efficiently monitor necroptosis in vitro in primary cells isolated from SMART Tg mice, such as macrophages and MEFs. Moreover, we could image cells in SMART Tg mice in vivo with two-photon excitation microscopy. We found that the FRET/CFP ratio in renal proximal tubular cells was elevated in cisplatin-treated mice compared to untreated mice. Taken together, these findings suggest that SMART Tg mice may be a valuable tool for visualizing necroptosis in vivo. It remains unclear whether we will be able to monitor necroptosis in all cells prepared from SMART Tg mice; nevertheless, SMART Tg mice may be useful for FRET analyses in various types of primary cells.

The FRET/CFP ratio was not increased in peritoneal macrophages from $Ripk3^{-/-}$ or $Mlkl^{-/-}$ mice following BV6/zVAD stimulation (in this study), and inducible expression of a constitutively active mutant of MLKL results in the increase in the FRET/CFP ratio in $Mlkl^{-/-}$ MEFs[26]. These results indicate that SMART responded to RIPK3-dependent conformational change of MLKL. Consistently, the averaged FRET/CFP ratios in renal tubular epithelial cells of cisplatin-treated SMART $Ripk3^{-/-}$ mice were comparable to those of untreated SMART $Ripk3^{-/-}$ mice. In sharp contrast, the averaged FRET/CFP ratios were higher in renal tubular epithelial cells of cisplatin-treated SMART $Mlkl^{-/-}$ mice than those of untreated SMART $Mlkl^{-/-}$ mice. These results implicated that, in addition to MLKL, SMART responded to, at least in part, conformational changes of RIPK3 under certain experimental conditions. Indeed, we also found that over-expressed SMART interacted with RIPK3 as well as MLKL in HEK293 cells. Further study will be required to determine which molecules, such as MLKL or RIPK3, are sensed by SMART depending on the different experimental conditions.

Considering organ accessibility from an external position, we decided to study the cisplatin-induced kidney injury model in SMART Tg mice. IHC revealed that pRIPK3+ cells appeared in the kidney on day 2 following cisplatin injection. This delayed kinetics of the appearance of necroptotic cells prevented the standard in vivo imaging from determining the FRET/CFP ratio before and after cisplatin stimulation. Therefore, we started imaging the FRET/CFP ratio of proximal tubular cells in the kidneys on day 2 following cisplatin injection. Although we expected that it would be difficult to capture cells undergoing necroptosis during short-term imaging (~2 h), we acquired images in SMART Tg mice that showed the increased FRET/CFP ratio in several renal proximal tubular cells. Of note, the increased FRET/CFP ratio in tubular cells usually occurred after the occlusion of proximal tubule lumens, suggesting that induction of necroptosis does not appear to be an initial event of tubular injury, but rather a late event following occlusion of proximal tubule lumens. As our present study did not investigate the causal relationship between the induction of the FRET and the occlusion of tubule lumens, we cannot formally exclude the possibility that these two events are independent, but occur simultaneously.

Of note, the numbers of cells showing the increased FRET/CFP ratios were relatively few, even in the occluded tubules of the cisplatin-treated kidneys of SMART Tg mice. This suggests that another type of cell death, such as apoptosis, ferroptosis, or necrosis, may simultaneously occur in cisplatin-induced kidney injury[45,46]. The mechanisms underlying cisplatin-induced necroptosis are still a matter of debate. A previous study reported that cisplatin-induced nephrotoxicity is mediated by TNF produced by renal parenchymal cells[47]. In contrast, a later study reported that combined treatment of renal tubular cells with TNF, TNF-like weak inducer of apoptosis (TWEAK), and IFNγ, but not TNF alone, results in necroptosis[43]. Moreover, under the conditions where expression of IAPs is attenuated by chemotherapeutic agents, such as etoposide and doxorubicin, the cell death-inducing signaling complex, termed the Ripoptosome, is spontaneously formed[48,49]. Notably, the formation of the Ripoptosome is independent of TNF, FasL, or TRAIL; therefore, the Ripoptosome is distinct from complex II, which originates from the death receptor-induced signaling complex. However, both complex II and the Ripoptosome comprise RIPK1, caspase 8, and FADD, evolving into the necrosome when caspase activity is attenuated[48,49]. Taken together, one might surmise that cisplatin treatment induces downregulation of the expression of IAPs that triggers the Ripoptosome formation, further increasing the susceptibility of renal proximal tubular cells to TNF- or TWEAK-induced apoptosis or necroptosis. Consistent with this

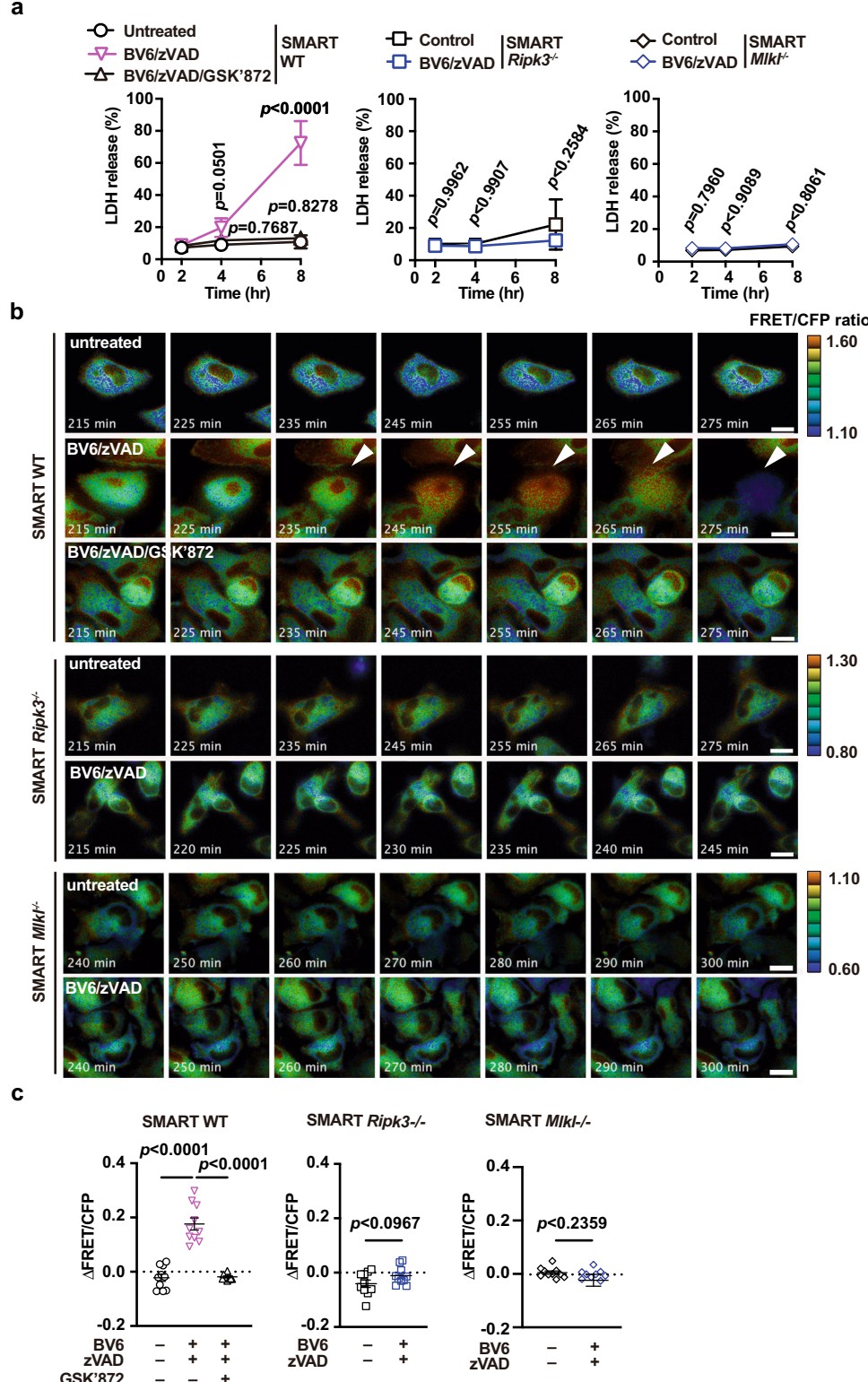

**Fig. 3 Crucial role for RIPK3 and MLKL in the FRET response of SMART in macrophages following necroptosis induction. a** Peritoneal macrophages from SMART WT, SMART *Ripk3*$^{-/-}$, SMART *Mlkl*$^{-/-}$ mice were untreated or stimulated with BV6 (1 μM) + zVAD (20 μM) or BV6 (1 μM) + zVAD (20 μM) + GSK'872 (5 μM) for the indicated periods. Cell death was determined with the LDH release assay. Results are mean ± SD of triplicate samples. **b**, **c** Peritoneal macrophages were stimulated and analyzed as described in **a**. Pseudocolored images show cellular changes in FRET/CFP ratio values in response to the indicated stimulations (**b**). FRET/CFP responses are color-coded according to the color scales (right). White arrowheads indicate cells undergoing necroptosis. Scale bars, 20 μm. Maximum changes detected in the FRET/CFP ratio (**c**). Results are mean ± SE (*n* = 10 cells). Each dot indicates an individual cell. Statistical analysis was performed with two-way ANOVA with Dunnett's multiple comparison test (**a**, left panel) or Sidak's multiple comparison test (**a**, middle and right panels), one-way ANOVA with Tukey's multiple comparison test (**c**, left panel), or the unpaired two-tailed Student *t*-test (**c**, middle and right panels). All results are representative of three independent experiments.

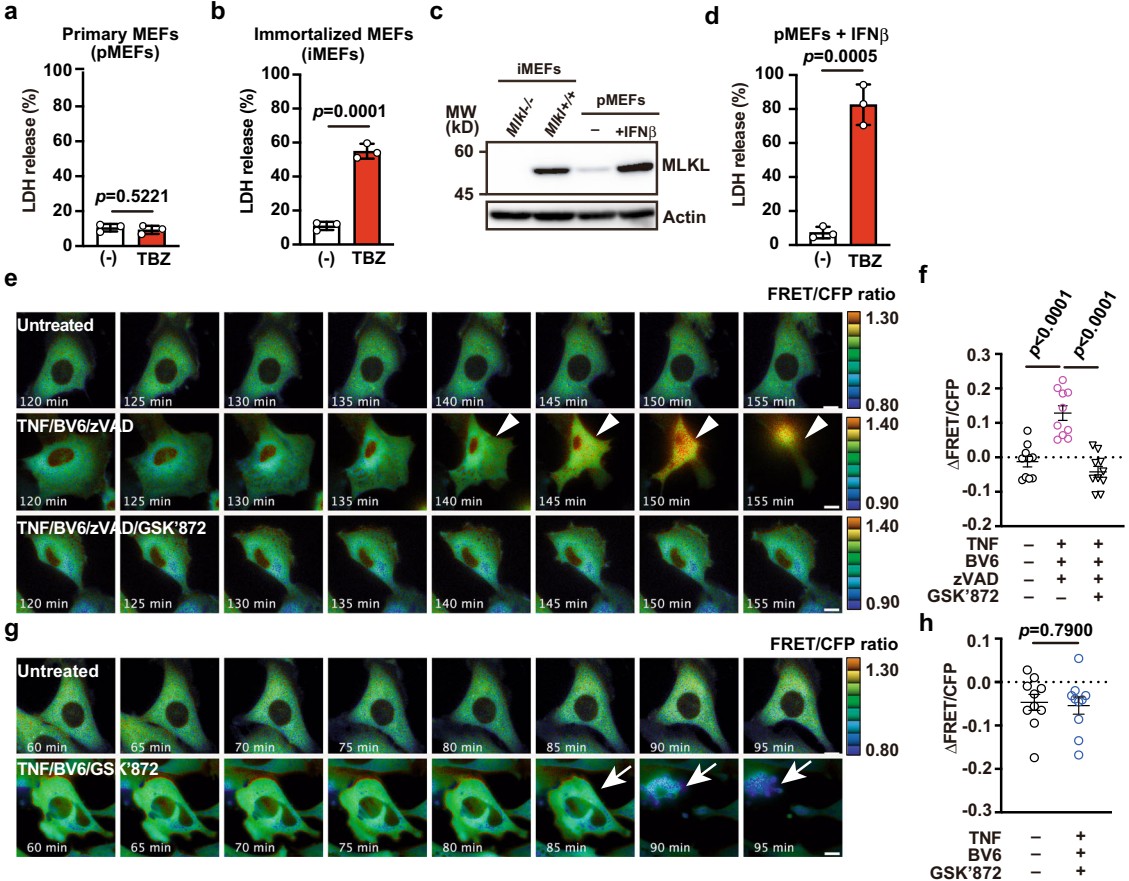

**Fig. 4 SMART responds to necroptosis, but not apoptosis, in MEFs. a**, **b** primary MEFs (pMEFs) (**a**) and immortalized MEFs (iMEFs) (**b**) were untreated or stimulated with TBZ (10 ng/mL TNF, 0.5 μM BV6, 20 μM zVAD) for 12 h. MEFs were generated from SMART Tg mice, and MEFs with less than ten passages were used as pMEFs. Cell death was determined with the LDH release assay. Results are mean ± SD of triplicate samples. **c** Western blots of lysates from untreated iMEFs from *Mlkl*$^{-/-}$ or wild-type mice, and untreated or IFNβ (2000 IU/mL)-treated pMEFs from SMART Tg mice. Blots were probed with anti-MLKL and anti-actin antibodies. **d** pMEFs were pretreated with IFNβ, and then stimulated with TBZ for 4 h. Cell viability was determined with the LDH release assay. Results are mean ± SD of triplicate samples. **e**–**h** pMEFs were untreated or stimulated with the indicated combination of agents (10 ng/mL TNF, 0.5 μM BV6, 20 μM zVAD, 5 μM GSK'872), and the FRET/CFP ratio was calculated. Pseudocolored images show cellular changes in the FRET/CFP ratio values in response to the indicated stimulations (**e**, **g**). FRET/CFP responses are color-coded according to the color scales (right). White arrowheads and white arrows indicate cells undergoing necroptosis (**e**) and apoptosis (**g**), respectively. Scale bars, 20 μm. Maximum changes of the FRET/CFP ratio are shown (**f**, **h**). Results mean ± SE (n = 10 cells per condition). Each dot indicates an individual cell. Statistical analyses were performed by the unpaired two-tailed Student's *t*-test (**a**, **b**, **d**, **h**) or one-way ANOVA with Tukey's multiple comparison test (**f**). All results are representative of at least two independent experiments.

idea, the RIPK1 inhibitor, Necrostain-1s, has been shown to suppress cisplatin-induced necroptosis[50]. Further study will be required to investigate this possibility.

FRET/CFP ratio elevations in renal tubular cells were not observed in SMART *Ripk3*$^{-/-}$ mice. That finding suggested that elevations in the FRET/CFP ratios depended on RIPK3 expression. In contrast to the previous studies[43,50], BUN and serum creatinine levels and numbers of CC3$^+$ cells were increased in *Ripk3*$^{-/-}$ mice compared to wild-type mice following cisplatin injection, at least under our experimental conditions. Although the mechanisms underlying the discrepancy between our and previous results are currently unknown, one can surmise that DAMPs released from necroptotic tubular cells may attenuate cisplatin-induced apoptosis.

Necroptotic cells may release large amounts of intracellular content into the extracellular space, resulting in various cellular responses[24]. Although we previously succeeded in simultaneously imaging the execution of necroptosis and the release of HMGB1 in vitro[26], it remains largely unknown where and when DAMPs are released from necroptotic cells in vivo. To address

this issue, it will be crucial to generate transgenic mice that express DAMPs fused to a fluorescent protein, such as HMGB1-mCherry. Then, a new mouse line could be generated by crossing SMART mice with HMGB1-mCherry mice. That model might allow the visualization of DAMP release and necroptosis simultaneously at single-cell resolution. In vivo imaging with SMART Tg mice could pave the way to a better understanding of necroptosis-induced biological responses in vivo.

Regarding limitations of the study, the FRET response of SMART was induced in renal proximal tubular cells in SMART *Mlkl*$^{-/-}$ mice following cisplatin injection. These results suggest that the FRET response of SMART may occur in certain cells lacking *Mlkl*, at least in vivo. Given that these cells do not die by necroptosis, we should also carefully interpret the results of the FRET response using SMART Tg mice in vivo.

## Methods
**Reagents**. We purchased the following reagents as follows: BV6 (B4653, ApexBio), cisplatin (AG-CR1-3590, Adipogen), GSK'872 (530389, Merck), IFNβ (12401-1,

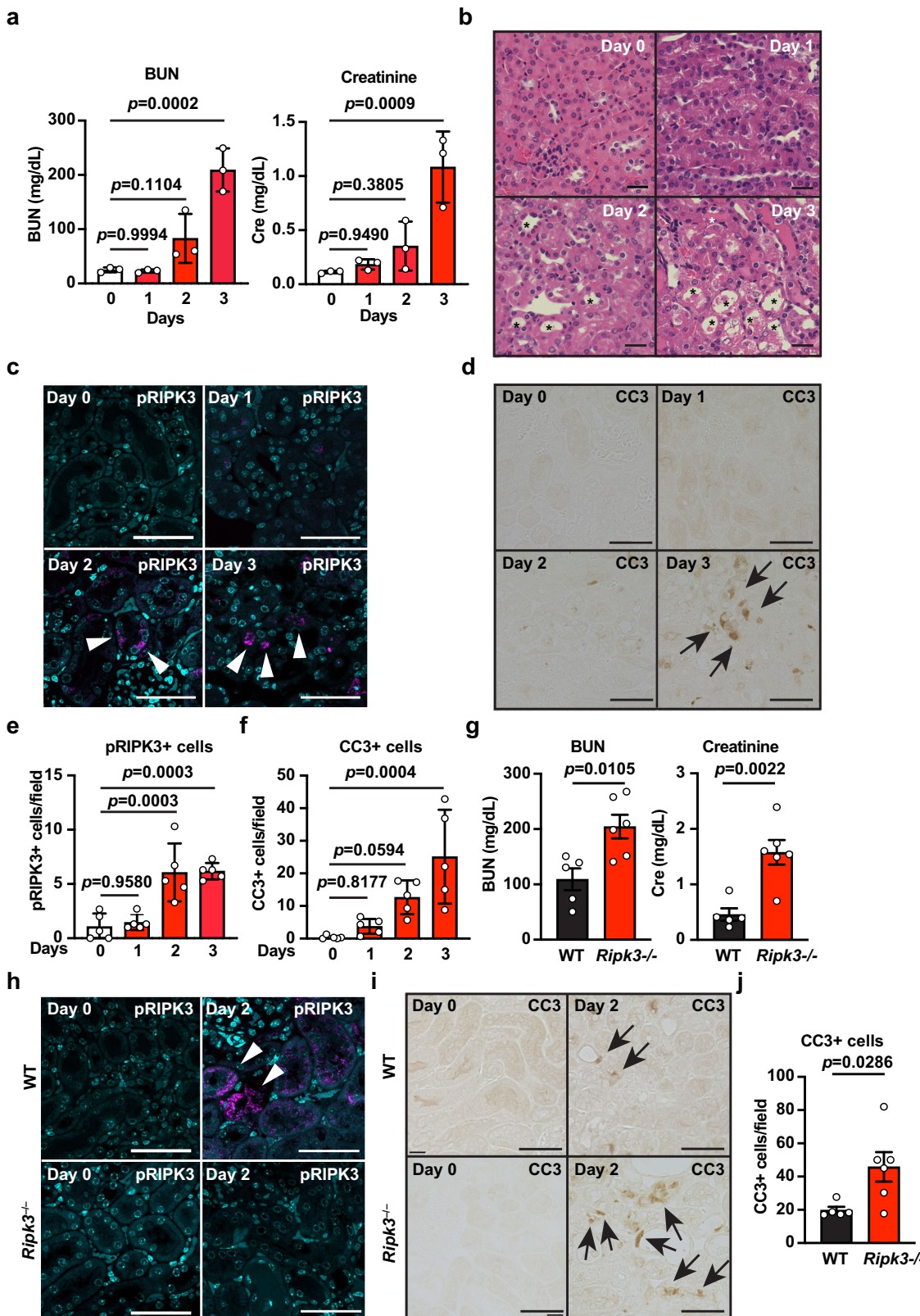

PBL-Assay Science), LPS (434, List Labs), nigericin (AG-CN2-0020, Adipogen), Murine TNF (PMC3013, Thermo Fisher Scientific), zVAD (3188-v, Peptide Institute), and 5z7Ox (499610, Calbiochem). The following antibodies were used in this study: anti-actin (A2066, Sigma-Aldrich, diluted 1:1000), anti-cleaved caspase 3 (9661, Cell Signaling Technology, diluted 1:1000), APC–conjugated anti-CD11b (20-0112, TOMBO Biosciences, diluted 1:100), PE-conjugated anti-F4/80 (50-4801, TOMBO biosciences, diluted 1:100), anti-FLAG (M2, Sigma-Aldrich, 1:1000), anti-green fluorescent protein (GFP) (sc-8334, Santa Cruz Biotechnology, diluted

1:5000), anti-MLKL (3H1, Millipore, diluted 1:1000), anti-Myc (9E10, Sigma-Aldrich, 1,000), anti-phospho-RIPK3 (57220, Cell Signaling Technology, diluted 1:1000), horseradish peroxidase (HRP)–conjugated sheep anti-mouse IgG (NA931, GE Healthcare, 1:5000), horseradish peroxidase (HRP)–conjugated donkey anti-rat IgG (712-035-153, Jackson ImmunoResearch Laboratories, Inc, 1:20000), HRP–conjugated donkey anti-rabbit IgG (NA934, GE Healthcare, 1:20000), biotin-conjugated donkey anti-rabbit IgG (E0432, Dako, 1:200), and Streptavidin-HRP (P0397, Dako, 1:300).

**Fig. 5 Administration of cisplatin induces necroptosis in renal proximal tubular cells. a–f** Eight-week-old wild-type were intravenously injected with cisplatin (20 mg/kg), then sacrificed at the indicated times. Concentrations of blood urea nitrogen (BUN) and serum creatinine were determined at the indicated times after cisplatin treatment (**a**). Results are mean ± SE ($n = 3$ mice per time point). Results are representative of two independent experiments. Kidney tissue sections were prepared at the indicated times after cisplatin treatment and stained with hematoxylin & eosin ($n = 5$ mice) (**b**). Scale bar, 50 μm. Asterisks indicate dilated proximal tubules. Kidney tissue sections were stained with anti-phospho-RIPK3 (pRIPK3) (**c**) or anti-cleaved caspase 3 (CC3) (**d**) antibodies. White arrowheads and black arrows indicate pRIPK3$^+$ and CC3$^+$ cells, respectively. Scale bars, 50 μm. Cell counts for pRIPK3$^+$ (**e**) and CC3$^+$ (**f**) cells, expressed as the number of positive cells per field. Results are mean ± SE ($n = 5$ mice). **g–j** Eight-week-old wild-type and $Ripk3^{-/-}$ mice were injected with cisplatin as in **a** and sacrificed on day 2 after injection (wild-type, $n = 5$ mice; $Ripk3^{-/-}$ mice, $n = 6$ mice). Concentrations of BUN and serum creatinine were determined as in **a** (**g**). Kidney tissue sections were stained with anti-pRIPK3 (**h**) or anti-CC3 (**i**) antibodies. Scale bars, 50 μm. The numbers of CC3$^+$ cells were counted as in **f** (**j**). White arrowheads and black arrows indicate pRIPK3$^+$ and CC3$^+$ cells, respectively. Statistical analyses were performed with one-way ANOVA and Dunnett's multiple comparison test (**a, e, f**) or the unpaired two-tailed Student $t$-test (**g, j**).

**Cell culture**. MEFs were prepared from mice of the indicated genotypes at E13.5–E14.5 after coitus using a standard method[51]. Briefly, a pregnant mouse was euthanized, and the uterus was removed from the abdomen. After removing the yolk sac and placenta from the embryo, the embryo was decapitated, and the limbs, tail, and internal organs were removed. The remaining body was minced with scissors and then incubated with trypsin/EDTA (0.25%)-PBS at 37 °C for 30 min. After adding 10% FBS-DMEM to cell suspensions to neutralize trypsin, cell suspensions were passed through a nylon mesh. Resultant cell suspensions were plated on a 10 cm dish and expanded until cells became confluent. Then cells were stocked in liquid nitrogen. MEFs below ten passages were used as primary MEFs (pMEFs) for experiments. Wild-type and $Mlkl^{-/-}$ MEFs were immortalized by transfection with a pEF321-T vector that encodes SV40 large T antigen[52]. HEK293 cells were obtained from ATCC. HEK293 cells and MEFs were maintained in a DMEM medium containing 10% fetal bovine serum (FBS).

**Mice**. $Ripk3^{-/-}$ mice[53] (provided by Genentech Inc.) and $Mlkl^{-/-}$ mice[54] (provided by Manolis Pasparakis) were described previously. C57BL/6 J mice (Sankyo Lab Service) were housed in a specific pathogen-free facility and received a routine chow diet and water $ad$ $libitum$. All animal experiments were performed according to the guidelines approved by the Institutional Animal Experiments Committee of Toho University School of Medicine (approval number: 21-54-400), Kyoto Graduate School of Medicine (approval number: Medkyo 21562), and the RIKEN Kobe branch (approval number: QA2013-04-11).

**Generation of SMART Tg mice**. SMART Tg mice were generated by microinjecting $Tol2$ mRNA and the pT2KXIG-mSMART vector into the cytoplasm of fertilized eggs from C57BL/6 N mice[30]. SMART $Ripk3^{-/-}$ mice and SMART $Mlkl^{-/-}$ were generated by crossing SMART Tg mice with $Ripk3^{-/-}$ mice and $Mlkl^{-/-}$ mice, respectively. Eight- to twelve-week-old male and female mice were used for the in vivo imaging.

**Western blotting**. Murine tissues were homogenized with a Polytron (Kinematica, Inc.) and lysed in RIPA buffer (50 mM Tris-HCl, [pH 8.0], 150 mM NaCl, 1% Nonidet P-40, 0.5% deoxycholate, 0.1% SDS, 25 mM β-glycerophosphate, 1 mM sodium orthovanadate, 1 mM sodium fluoride, 1 mM phenylmethylsulfonyl fluoride, 1 μg/ml aprotinin, 1 μg/ml leupeptin, and 1 μg/ml pepstatin). After centrifugation, cell lysates were subjected to SDS polyacrylamide gel electrophoresis and transferred onto polyvinylidene difluoride membranes (IPVH 00010, Millipore). The membranes were immunoblotted with the indicated antibodies and developed with Immobilon Western Chemiluminescent HRP Substrate (WBKLS0500, Millipore). The signals were analyzed with an Amersham Imager 600 (GE Healthcare Life Sciences). Uncropped images of all blots in Figs. 1, 4 and Supplementary Fig. 3 are included in Supplementary Data 1.

**Transient transfection and co-immunoprecipitation**. HEK293 cells ($1.5 \times 10^6$ cells/ 60-mm-dish) were transiently transfected with expression vectors for SMART and Myc-tagged MLKL or FLAG-tagged RIPK3 with PEI MAX 40000 (24765, Polyscience). The expression vector for SMART was previously described[26]. Expression vectors for murine MLKL and murine RIPK3 were constructed as follows: Briefly, the cDNA of murine $Mlkl$ was amplified by RT-PCR with the primers (forward, 5′-CTAAATGGGGAATTCATGGATAAATTGGGACAGAT-3′ and reverse, 5′-AGATGCATGCTCGAGTTACACCTTCTTGTCCGTGG-3′) using mRNA of the murine ovary as a template. cDNA of murine $Ripk3$ was amplified by PCR with the primers (forward, 5′-AAAAAGAATTCATGTCTTCTGTCAA GTTATGGCCT-3′ and reverse, 5′-TTTTTCTCGAGCTACTTGTGGAAGGGC TGCCAGCC-3′) using pGEM-mRIPK3 as a template that was purchased from Sino Biological Inc. (MG51069-G). PCR-amplified products of these genes were subcloned into pcDNA3-6xMyc and pCR3-2xFLAG vectors, resulting in the generation of pcDNA3-6xMyc-mMLKL and pCR3-2xFLAG-mRIPK3, respectively.

Twenty-four hours after transfection, cells were lysed with IP buffer (50 mM Tris-HCl [pH 8.0], 150 mM NaCl, 0.5% Nonidet-40, 25 mM β-glycerophosphate, 1 mM sodium orthovanadate, 1 mM sodium fluoride, 1 mM phenylmethylsulfonyl fluoride, 1 μg/ml aprotinin, 1 μg/ml leupeptin, and 1 μg/ml pepstatin). After centrifugation, the supernatants were immunoprecipitated with GST protein or GST-fused anti-GFP nanobody[55] adsorbed to glutathione–Sepharose 4B beads (17075601, GE Healthcare Life Science) at 4 °C for overnight. Approximately 0.5 μg of GST or GST-fused anti-GFP nanobody were used for immunoprecipitation. Immunoprecipitated proteins were subjected to SDS-PAGE and analyzed by immunoblotting as described above. The expression of transfected proteins was verified by immunoblotting with total cell lysates.

Induction and purification of GST-fused anti-GFP nanobody were performed according to the original method with minor modification[56]. $E$. $coli$ BL21(DE3) was transformed with the expression vector for GST-fused anti-GFP nanobody (Addgene 61838). The transformed bacteria were incubated in 2 ml of LB medium containing 50 μg/ml ampicillin at 37 °C overnight, followed by inoculation in 200 ml of LB medium, and the culture was continued at 37 °C. When the bacteria reached 1.0 OD$_{600}$, IPTG was added to a final concentration of 0.1 mM and incubated at 20 °C for 24 h to induce protein expression. Then, the bacteria were harvested and resuspended in binding buffer (PBS, 1% Triton-X, 5 mM DTT, 1 mM phenylmethylsulfonyl fluoride, 1 μg/ml aprotinin, 1 μg/ml leupeptin, and 1 μg/ml pepstatin), followed by sonication by a Polytron. After centrifugation, the supernatants were incubated with 1 ml of glutathione–Sepharose 4B beads (1:1 slurry) for 2 h. Then, the beads were washed with wash buffer (PBS, 0.1% Triton-X, 5 mM DTT) nine times, and then the volume of the beads was adjusted to 1 ml with wash buffer.

**Flow cytometry**. Cells were stained with anti-CD11b and anti-F4/80 antibodies in PBS containing 2% FBS. The prepared cells were gated on forward and side scatters to identify the lymphocyte population and then to discriminate doublets. Cells were analyzed with a BD FACSCant II flow cytometer (BD Biosciences) and FlowJo software (BD Biosciences).

**Cell death assay**. Macrophages were plated onto 96-well plates and cultured for 12 h in RPMI medium containing 10% FBS. Macrophages were stimulated with the IAP antagonist, BV6 (1 μM), in the presence of the apoptosis inhibitor, zVAD (20 μM), the RIPK3 inhibitor, GSK'872 (5 μM), or both, as indicated, for the times indicated. To induce pyroptosis, macrophages were pretreated with LPS (10 ng/mL) for 4 h, then stimulated with nigericin (10 μM) for the indicated times. Primary MEFs were untreated or pretreated with IFNβ (2000 IU/mL) for 24 h, then stimulated with TNF (10 ng/mL) and BV6 in the presence of zVAD (20 μM) or GSK'872 (5 μM), as indicated, for the times indicated. The concentrations of LDH released from cells were determined with a Cytotoxicity Detection Kit (11644793001, Roche) according to the manufacturer's instruction[57].

In some experiments, macrophages were pretreated with LPS (10 ng/mL) for 4 h and then stimulated with 5z7Ox (125 nM) in the presence of zVAD (20 μM), GSK'872 (5 μM), or both, as indicated, for 4 h. Cell viability was determined with a water-soluble tetrazolium salts assay (Cell Count Reagent SF 07553-44, Nacalai Tesque).

**FRET analysis in vitro**. FRET analysis was performed according to the previously published method with minor modification[26]. Briefly, FRET signals were imaged with a DeltaVision microscope system (GE Healthcare) built on an Olympus IX-71 inverted microscope base equipped with a Photometric Coolsnap HQ2 CCD camera and a 60×/NA1.516 PlanApo oil immersion lens (Olympus). For live-cell imaging FRET sensors, cells were seeded on gelatin-coated CELLview Cell Culture Dishes (Greiner Bio-One) and maintained in an incubator at 37 °C with 5% CO$_2$. For imaging, cells were observed with a Blue excitation filter (400-454 nm), two emission filters (blue-green, 463–487 nm for ECFP; yellow-green, 537–559 nm for Ypet), and a C-Y-m polychronic mirror. The FRET emission ratio (FRET/CFP) was calculated with SoftWoRx (Applied Precision Inc) by dividing the excitation at 436 nm and emission at 560 nm (FRET) by the excitation at 436 nm and emission at 470 nm (CFP). For statistical analyses, the obtained images were analyzed with ImageJ and MetaMorph software. The ΔFRET/CFP ratios were calculated by

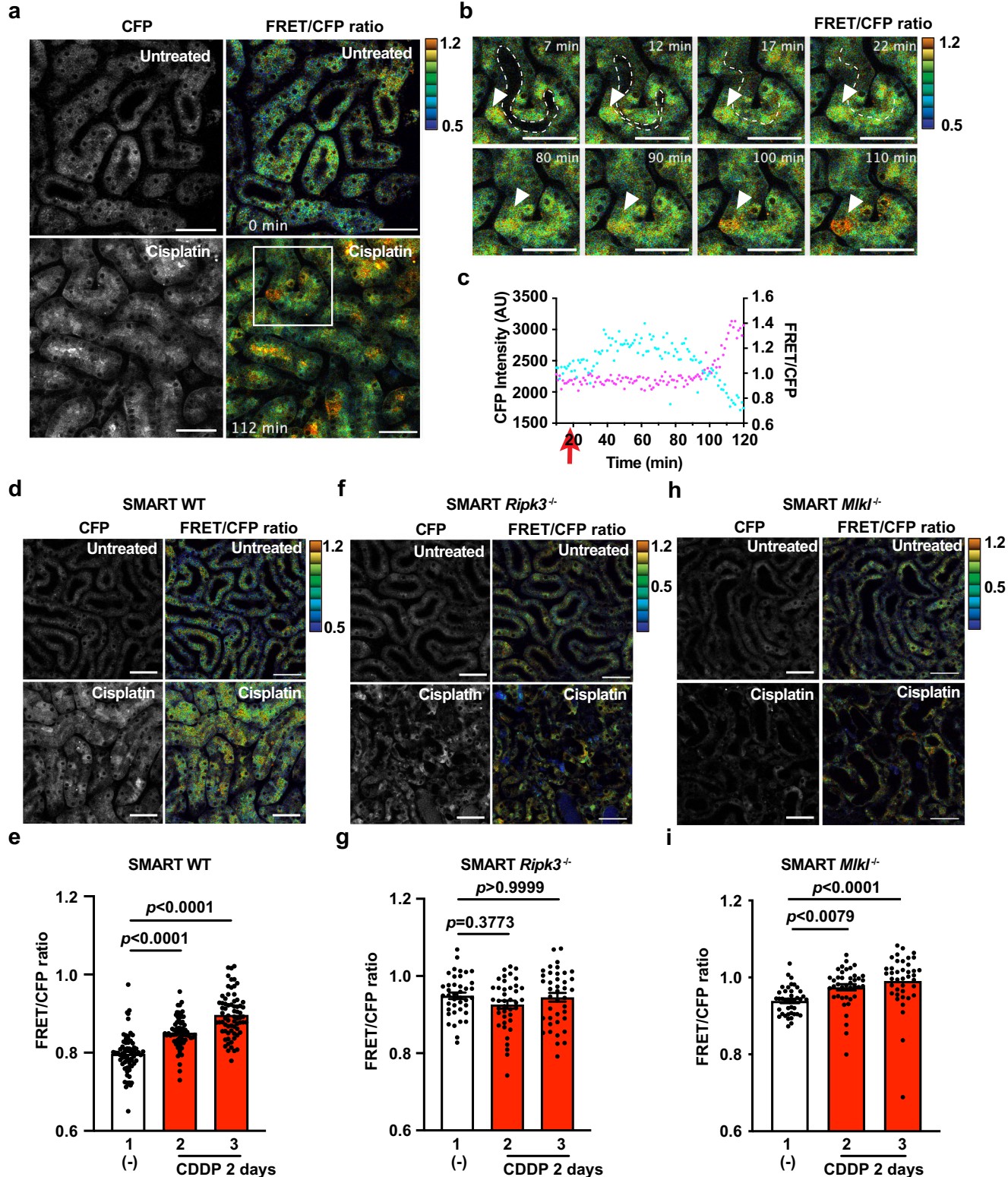

COMMUNICATIONS BIOLOGY | (2022)5:1331 | https://doi.org/10.1038/s42003-022-04300-0 | www.nature.com/commsbio

subtracting the FRET/CFP ratio at time 0 from the FRET/CFP ratio at the indicated times.

**Isolation of peritoneal exudate macrophages**. To isolate peritoneal exudate macrophages, 8- to 12-week-old mice of the indicated genotypes were intraperitoneally injected with 2.5 ml 3% thioglycollate (T9032, Sigma). On day 4 after the thioglycollate injection, anesthetized mice were intraperitoneally injected with ice-cold PBS; then, the peritoneal cells were harvested when the PBS was recovered. This procedure was repeated twice. The peritoneal cells were collected by

centrifugation and resuspended in ACK lysing buffer (0.15 M NH$_4$Cl, 10 mM KHCO$_3$, 0.1 mM EDTA, pH7.2) for 1 min at room temperature to remove erythrocytes. Harvested cells were placed in plates with RPMI medium. After removing non-adherent cells, the remaining cells were analyzed by flow cytometry and used as primarily peritoneal macrophages. Approximately 80% of these cells were positively stained with CD11b and F4/80 antibodies.

**TNF-induced SIRS model of mice**. Eight- to twelve-week-old mice of the indicated genotypes were injected with mTNF (0.5 mg/kg in PBS) via the tail vein.

**Fig. 6 Crucial role for RIPK3 in the FRET response of SMART in renal proximal tubular cells after cisplatin administration. a–c** Eight- to twelve-week-old SMART Tg mice were untreated or treated with 20 mg/kg cisplatin. On day 2, the left-side kidneys of untreated and cisplatin-treated mice were analyzed with two-photon excitation microscopy. Black-and-white and pseudocolored images show CFP intensities and the FRET/CFP ratios, respectively, in kidneys of untreated (upper panel) or cisplatin-treated (lower panel) wild-type mice (**a**). FRET/CFP responses are color-coded according to the color scales. Scale bars, 50 μm. Magnified images of the areas outlined in white boxes (**b**). Dotted white lines indicate proximal tubule lumens. White arrowheads indicate cells that show increases in the FRET/CFP ratios. FRET/CFP responses are color-coded according to the color scales. Scale bars, 50 μm. Graphs show CFP intensities (cyan) and the FRET/CFP ratios (magenta) of the cells indicated in **b**, calculated at times indicated (**c**). Red arrows indicate the time that the tubule lumens were occluded. **d–g** Eight- to twelve-week-old SMART WT (**d**), SMART Ripk3$^{-/-}$ (**f**), or SMART Mlkl$^{-/-}$ (**h**) mice were untreated or treated with cisplatin, as described in **a**. FRET/CFP responses are color-coded according to the color scales. Scale bars, 50 μm. Graphs show the averaged FRET/CFP ratios observed in tubules from untreated or cisplatin-treated SMART WT (**e**), SMART Ripk3$^{-/-}$ (**g**), SMART Mlkl$^{-/-}$ (**i**) mice. Results are mean ± SE (n = 70 tubules in SMART WT mice; n = 40 tubules in SMART Ripk3$^{-/-}$ mice; n = 40 tubules in SMART Mlkl$^{-/-}$ mice). Each number indicates an individual mouse. Results are representative of two independent experiments. Statistical analyses were performed by one-way ANOVA with Dunnett's multiple comparison test.

Injected mice were sacrificed by cervical dislocation at 3 h, and ileal tissues were used for subsequent analysis.

**Injection of cisplatin into mice**. To produce cisplatin-induced kidney injuries, 8- to 12-week-old mice of the indicated genotypes were injected with cisplatin (20 mg/ kg in PBS). Injected mice were sacrificed at the indicated times, and sera and kidney tissues were collected for subsequent analyses.

**Histological, immunohistochemical, and immunofluorescence analyses**. Tissues were fixed in 10% formalin and embedded in paraffin blocks. Paraffin-embedded kidney sections were used for hematoxylin and eosin staining, immunohistochemistry, and immunofluorescence analyses. For immunohistochemistry, paraffin-embedded sections were treated with Instant Citrate Buffer Solution (RM-102C, LSI Medicine) for antigen retrieval. Next, tissue sections were stained with an anti-CC3 antibody, followed by the secondary antibody, a biotin-conjugated anti-rabbit antibody. Streptavidin-HRP was added for visualization. Images were acquired with an all-in-one microscope (BZ-X700, Keyence) and analyzed with a BZ-X Analyzer (Keyence). CC3$^+$ cells were automatically counted in three randomly selected high-power fields (original magnification, ×40) per kidney, with a Hybrid Cell Count (Keyence).

For the immunofluorescence analyses, tissue sections were preincubated with MaxBlock$^{TM}$ Autofluorescence Reducing Kit (MaxVision Biosciences), according to the manufacturer's instructions. Next, tissue sections were stained with anti-pRIPK3 antibody, followed by visualization with the tyramide signal amplification method, according to the manufacturer's instructions (NEL741001KT, Kiko-tech). Images were acquired with an LSM 880 (Zeiss). The images were processed and analyzed with ZEN software (Zeiss) and an image-processing package, Fiji (https:// fiji.sc/). pRIPK3$^+$ cells were counted manually.

**Measurement of blood urea nitrogen and creatinine**. After the cisplatin injection, serum samples were collected on days 0, 1, 2, and 3. Serum creatinine (measured with an enzymatic method) and blood urea nitrogen (measured with the urease-glutamate dehydrogenase method) were measured by Oriental Yeast Co., Ltd.

**Imaging cisplatin-induced kidney injury**. Living mice were observed with an FV1200MPE-BX61WI upright microscope (Olympus) equipped with an XLPLN 25XW-MP 25X/1.05 water-immersion objective lens (Olympus). The microscope was equipped with an InSight DeepSee Ultrafast laser (0.95 Watt, 900 nm; Spectra-Physics, Mountain View, CA). The scan speed was set at 2 μs/pixel. The excitation wavelength for CFP was 840 nm. Fluorescence images were acquired with the following filters and mirrors: an infrared-cut filter (BA685RIF-3); two dichroic mirrors (DM505 and DM570); and three emission filters, including an FF01-425/ 30 (Semrock, Rochester, NY) for the second harmonic generation; a BA460-500 (Olympus) for CFP; and a BA520-560 (Olympus) for FRET. The microscope was also equipped with a two-channel GaAsP detector unit and two multi-alkali detectors. FluoView software (Olympus) was used to control the microscope and acquire images. Acquired images were saved in the multilayer 12-bit tagged image file format and processed and analyzed with Metamorph software.

Intravital mouse imaging was performed according to the previously published method with minor modification[58]. Briefly, 2 days before imaging, cisplatin was injected intravenously. Then, mice were anesthetized with isoflurane (1.5% inhalation, 0.5 L/min). To observe the kidney, the mouse was placed in the prone position on an electric heat pad maintained at 37 °C. A 1-cm incision was made in the skin of the lower back and underlying peritoneum to expose ~0.25 cm$^2$ of tissue. The exposed tissue was imaged with an aspiration fixation system. The obtained images were analyzed by Metamorph. The FRET/CFP ratios of relevant areas were calculated with ImageJ software. Then, the averaged FRET/CFP ratios were calculated for each proximal tubule.

**Imaging of the ileum of TNF-induced SIRS injury**. Mice were anesthetized as described above. An incision was made in the median line of the abdominal wall. The ileum was pulled out of the abdominal cavity and filled with phosphate-buffered saline (PBS) to minimize peristalsis. The serosa of the ileum was attached to the cover glass of the objective lens and observed with the inverted two-photon excitation microscope as described above. For the stimulation of TNF, we injected mice with mTNF (0.5 mg/kg in PBS) via the tail vein.

**Statistics and reproducibility**. Statistical analyses were performed with the unpaired two-tailed Student's t-test or one-way or two-way ANOVA with Dunnett's, Sidak's, or Tukey's multiple comparison test, as appropriate. P values <0.05 were considered statistically significant. All experiments were performed at least twice to confirm reproducibility.

## Data availability

The authors declare that the data supporting this study are available within the paper and its supplementary movies. Source data behind the graphs in Figs. 1–6 are available as Supplementary Data 1. Uncropped images of all gels/blots in Figs. 1, 4 and Supplementary Fig. 3 are included in Supplementary Data 1. Other datasets generated during and/or analyzed during the current study are available from the corresponding author upon reasonable request.

## Materials availability

All the biological materials, including SMART Tg mice used in this study, are available from the corresponding authors upon reasonable request. Obtaining Ripk3$^{-/-}$ and Mlkl$^{-/-}$ mice requires Material Transfer Agreement (MTA) from Genentech Inc. and Manolis Pasparakis, respectively.

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

## Acknowledgements

We thank K. Asano for technical advice, Genentech Inc. for *Ripk3*$^{-/-}$ mice, M. Pasparakis for *Mlkl*$^{-/-}$ mice, and K. Kawakami for the transposon-based expression vector system. This work was supported, in part, by Grants-in-Aid for Scientific Research (B) 20H03475 (to H.N.), Scientific Research (C) 19K07399 (to K.M.), and Scientific Research (C) 20K05238 (to S.M.) from the Japan Society for the Promotion of Science (JSPS), a Grant-in-Aid for Scientific Research on Innovative Areas (JP16H06280) —Platforms for Advanced Technologies and Research Resources "Advanced Bioimaging Support", from the Japan Agency for Medical Research and Development (AMED), under Grant Numbers 21gm1210002 (to H.N.) and 21wm0325050 (to K.M.), grants from the Ministry of Education, Culture, Sports, Science, and Technology, Japan, Toho University Grant for Research Initiative Program (TUGRIP) (to H.N.), the Science Research Promotion Fund, and The Promotion and Mutual Aid Corporation for Private Schools of Japan (to H.N.), a GSK Japan Research Grant 2020 (to K.M.), and a grant from the Takeda Science Foundation (to K.M.).

## Author contributions

S.M., K.M., Kenta T., M.M., and H.N. designed research; S.M., Kanako T., S.K.-S., and T.S. performed research; K.S., K.A., and M.O. contributed to new reagents/analytical tools; S.M., Kanako T., K.M., Kenta T., Y.Y., T.M., M.M., and H.N. analyzed data; S.M. and H.N. wrote the manuscript.

## Competing interests

The authors declare no competing interests.

## Additional information

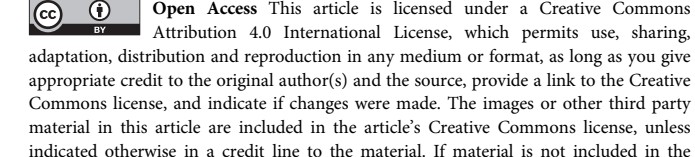

