## [Peer Review File · Communications Biology]

Reviewers' comments:

Reviewer #1 (Remarks to the Author):

Murai et al. describe the application of their previously reported MLKL-activation reporter, termed SMART, in vivo in an acute kidney injury model. This is very exciting science and is well presented. There is a lot to like about this study – the authors show that only some cell types in the kidney undergo necroptosis on cisplatin administration, and could validate the biosensor as not detecting other death modes in the kidney in line with their earlier cellular studies. The focus on detecting necroptosis activation is on RIPK3 phosphorylation and uses what appears to be suitable controls to validate the staining. However the authors should provide a stronger validation for why pRIPK3 and not MLKL as the readout considering the earlier report that MLKL oligomerization was the step detected by the biosensor. I have some additional queries, which I hope the authors consider addressing in revision.

Further points for consideration:

1. In the discussion, the authors mention that TNF SIRS was first tested but the data were not shown in the manuscript. I feel the authors need to show the data or at least explain why they are not. I do not feel it is appropriate to draw conclusions from data not presented (line 257-259).
2. On page 11, the authors should provide an explanation for why GSK872 was required to promote apoptosis. It is more typical to rely solely on TNF/BV6. Is this a quirk of the iMEFs used?
3. The activation mechanism of the SMART sensor in Figure 1a is difficult to understand. It is not clear what the component domains shown in the schematic are. Nor is it clear why it is RIPK3 based. In the first report, it was MLKL oligomerization that was reported to be the focus of the detector, rather than pRIPK3.
4. Relatedly, much of what we know about MLKL's activation mechanism has been reported in recent literature, which is strangely not cited. It would be of interest to the broad readership of Comms Biol to have greater context for how MLKL is activated by RIPK3 and therefore where in the pathway SMART detects activation. Additionally a citation is required to support the idea that necroptosis is not a developmental pathway on line 251.
5. Almost all figures: the microscopy images have a colour scale on the right but it is not labelled within the figure as to what this represents.
6. Figure 1b. It would more useful to label the blot with what was blotted with. There is no SMART antibody so it'd be useful to identify how SMART expression was being detected. It doesn't appear to be MLKL because there's no endogenous band in WT samples.
7. In figure 5e there are multiple comparisons between conditions for which significance is shown. It is not clear what the significant samples are being compared to. Bars should be put above the plots to show what 2 samples are being compared and whether the difference is significant.
8. Some more details required in Methods. The source of MLKLko MEFs and a citation to allow others to repeat the study as required; are the mice N or J background?; why 20mg/kg cisplatin? The data to find this as an optimal concentration are not shown and I feel would be useful for the readership; Line 439 – what anaesthetic was used?

Minor

Some plurals are incorrect throughout, e.g. findings on line 160 and conditions on line 155; period on lines 223 and 231 rather than periods. I'm sure these will be easily corrected in proof/copyediting

however. A typo of dilation on line 197.

Line 263 – delete relatively as it is unclear what it is relative to

Line 287 – it would be constructive for readers if the authors could specify how the ripoptosome differs from the necrosome which is more conventionally associated with necroptosis.

Line 293. Please specify the RIP1 inhibitor. Not all operate via the same mode of action

Line 296. RIPK3^{-/-} should have superscript ^{-/-}

Reviewer #2 (Remarks to the Author):

In their paper Shin Murai et al., extended their model of measuring RIPK3 activity from in-vitro into in-vivo model by introducing their FRET biosensor, SMART into mice to generate a reporter mice. This is very important tool, which is very much needed, for the growing research on necroptosis. In general the data and methodologies during the manuscript are appropriate and are clearly presented. There is one major issue with the data throughout the all manuscript, which is the claim that this is a specific system to monitor only necroptosis. Although, they have nicely showed that it can detect necroptosis and not other cell death mechanisms as apoptosis nor pyroptosis, they did not show that it is detecting necroptosis. In fact, their system detect the ability of phosphorylated RIPK3 (pRIPK3) to bind the binding domain of the necroptotic executor MLKL. This is very important as for example, if they will repeat their experiments using an MLKL KO mice or cells, they will probably detect necroptosis, which of course will be wrong, as necroptosis can't be executed without MLKL. Thus, in cells that don't express sufficient MLKL this system may results in false positive result, as it will measure RIPK3 activity. if necroptosis is

Therefore, I will suggest that the authors will correct their text (including title and abstract) to clearly state that this system monitor RIPK3. They should discuss that as necroptosis is known, as for today, to be the main downstream pathway of pRIPK3 oligomeraization, there system is can be used to analyze necroptosis, but it should be confirmed with other methods as specifically looking on MLKL phosphorylation or using MLKL KO.

Reviewer #3 (Remarks to the Author):

In the presented manuscript, Murai et al. present their work on a FRET biosensor to allow necroptosis assessment in kidney tubular cells and in a mouse model of cisplatin-induced AKI. They refer to their technology as sensor for MLKL activation based on FRET "SMART" – this was published in NatComm in 2018 - and now generated SMART transgenic mice. They now investigated these mice with what they refer to as live cell imaging for secretion (LCI-S) of DAMPs. That latter system is based on a sandwich ELISA and allows them to detect HMGB1-release in parallel to necroptosis monitoring. This technology will be very helpful to many laboratories. More importantly, however, this manuscript refers adds important evidence to the role of necroptosis in acute kidney injury, and in particular in tubular cells. Authors should point out that so far, necroptosis was detected in human biopsy sampels from AKI patients only by immunohistochemistry (Gong et al., Cell, 2017, PMID 28388412) which is hard to control in humans. Adding this novel assay convincingly supports the initial notion of necroptosis in tubular cells. Not only that, maybe this technology exhibits the most convincing piece of evidence for necroptosis involvement in AKI.

That said, I have a couple of very minor concerns regarding this manuscript that in my understanding must not be held back from the scientific community for any reason. It should be published as soon as possible, and the tool should be made available to scientists working in this field.

Minor concerns

- 1) Reference #36 is problematic as the same group has toned down their conclusion of this paper in a subsequent manuscript. Especially when citing this manuscript for the effects observed by RIPK kinase inhibitors, I recommend caution.
- 2) The occurrence of secondary apoptotic cells in Fig. 5d might be best interpreted as a regenerative reaction that only comes in at day 3 following cisplatin. Inhibition of apoptosis in this model does not result in protection from the increase in BUN or serum creatinine concentrations.
- 3) The intravital imaging is fantastic, congratulations!!!

Reviewers' comments:

Reviewer #1 (Remarks to the Author):

Murai et al. describe the application of their previously reported MLKL-activation reporter, termed SMART, in vivo in an acute kidney injury model. This is very exciting science and is well presented. There is a lot to like about this study – the authors show that only some cell types in the kidney undergo necroptosis on cisplatin administration, and could validate the biosensor as not detecting other death modes in the kidney in line with their earlier cellular studies. The focus on detecting necroptosis activation is on RIPK3 phosphorylation and uses what appears to be suitable controls to validate the staining. However the authors should provide a stronger validation for why pRIPK3 and not MLKL as the readout considering the earlier report that MLKL oligomerization was the step detected by the biosensor. I have some additional queries, which I hope the authors consider addressing in revision.

RESPONSE: Thank you for the positive comments on our manuscript that encourage us to submit the revised manuscript. Regarding the molecules sensed by SMART, we apologize that we did not describe the mechanisms clearly enough. We have now described the findings more clearly and in more detail. We have also toned down some descriptions, such as “SMART senses necroptosis”, because we have only showed that SMART sensed conformational changes of MLKL, but not necroptosis itself. Please see the following responses.

Further points for consideration:

1. In the discussion, the authors mention that TNF SIRS was first tested but the data were not shown in the manuscript. I feel the authors need to show the data or at least explain why they are not. I do not feel it is appropriate to draw conclusions from data not presented (line 257-259).

RESPONSE: Thank you for pointing out a critical issue. We tried several times to visualize the FRET response of SMART using a TNF-induced SIRS model. However, it has not been easy to obtain consistent results due to the following

reasons. Thus, we have concluded that this model was not suitable for visualizing the FRET response of SMART by two-photon excitation microscopy.

- 1) First, as mentioned in the revised manuscript, only small numbers of intestinal epithelial cells expressed phospho-RIPK3 (a hallmark of necroptosis) in wild-type mice following high doses of TNF injection (0.5 mg/kg) (Page 13, lines 208-209 and Supplementary Figure. 2a).
- 2) Second, it was difficult to measure the FRET/CFP ratio of SMART in an individual epithelial cell continuously with two-photon excitation microscopy due to constant peristalsis of the small intestine (Supplementary Movie 4).
- 3) Third, we usually ligated 2 cm long ileum at the rostral and caudal sides and filled the ligated ileum with PBS to reduce peristalsis. This ligation gradually induced ischemia, resulting in an increased FRET/CFP ratio of many epithelial cells at later time points after TNF injection (Supplementary Figure 2b, c, and Supplementary Movie 4).

We have included these results in Supplementary Figure. 2 and new Supplementary Movie 4 and mentioned these results in the revised manuscript (page 13, lines 204-212).

Accordingly, old Supplementary movies 4 and 5 were changed to new Supplementary movies 5 and 6 in the revised manuscript, respectively.

2. On page 11, the authors should provide an explanation for why GSK872 was required to promote apoptosis. It is more typical to rely solely on TNF/BV6. Is this a quirk of the iMEFs used?

RESPONSE: As suggested, TNF/BV6 alone may be sufficient to induce apoptosis in wild-type MEFs. Nevertheless, we used GSK'872 to block necroptosis completely.

3. The activation mechanism of the SMART sensor in Figure 1a is difficult to understand. It is not clear what the component domains shown in the schematic are. Nor is it clear why it is RIPK3 based. In the first report, it was MLKL oligomerization that was reported to be the focus of the detector, rather than pRIPK3.

We apologize that we did not describe the mechanisms clearly enough. Regarding the molecules sensed by SMART, we previously reported that oligomers of MLKL increase the FRET/CFP ratio of SMART in several cell lines (Murai et al., 2018). However, in the initially-submitted manuscript, we only analyzed SMART *Ripk3*^{-/-} mice, but not SMART *Mlkl*^{-/-} mice. Therefore, we only stressed that phosphorylation of RIPK3 was important for the increase in the FRET/CFP ratio in primary macrophages and renal tubular epithelial cells. To substantiate our previous model (Murai et al., 2018), we generated SMART *Mlkl*^{-/-} mice and analyzed peritoneal macrophages and renal tubular epithelial cells. The increase in the FRET/CFP ratio was abolished in BV6/zVAD-treated macrophages from SMART *Mlkl*^{-/-} mice (revised Figure 3a, b, c). Surprisingly, the averaged FRET/CFP ratios were higher in renal tubular epithelial cells of cisplatin-treated SMART *Mlkl*^{-/-} mice than those of untreated SMART *Mlkl*^{-/-} mice (revised Figure 6h, i). Related to this observation, we found that overexpressed SMART interacted with MLKL and RIPK3 in HEK293 cells (new Supplementary Figure 3). These results suggest that SMART may have the ability, at least in part, to sense conformational changes of RIPK3 through the interaction with RIPK3 under *Mlkl*-deficient cells. However, considering our previous results (Murai et al., 2018) and the present results (revised Figures 1 to 3), we assume that SMART mainly senses conformational changes of MLKL by interacting with MLKL under *Mlkl*-sufficient cells. Accordingly, we have revised Figure 1a to include the current our model and Figures 3 and 6h, i to include new results using SMART *Mlkl*^{-/-} mice. In addition, we have made new Supplementary Figure 3 to support the results using cisplatin-treated SMART *Mlkl*^{-/-} mice. We have mentioned these in the revised manuscript (Page 3, lines 42, 44, Pages 6 to 7, lines 100-109, Page 8, lines 116-119, Pages 11 to 12, lines 171-184, Page 16, lines 256-264, Pages 17-18, lines 276-290, Page 41, lines 777-782).

As suggested by reviewer 2, we have changed the title to “Generation of transgenic mice expressing a FRET biosensor, SMART, that responds to necroptosis”.

4. Relatedly, much of what we know about MLKL’s activation mechanism has been reported in recent literature, which is strangely not cited. It would be of interest to the broad readership of Comms Biol to have

greater context for how MLKL is activated by RIPK3 and therefore where in the pathway SMART detects activation. Additionally a citation is required to support the idea that necroptosis is not a developmental pathway on line 251.

RESPONSE: Sorry for the insufficient citation of the papers to explain the mechanisms of how RIPK3 activates MLKL. We have mentioned the mechanisms underlying RIPK3-dependent activation of MLKL by citing several references (Page 5, lines 71-79).

Regarding the role for necroptosis in normal development, we have included the references, which showed that necroptosis was dispensable for normal murine development. We have moved the paragraph to the Result section (Page 13, lines 202-204).

5. Almost all figures: the microscopy images have a colour scale on the right but it is not labelled within the figure as to what this represents.

RESPONSE: As suggested, we have added the information that color scales indicated the FRET/CFP ratio in the revised Figures.

6. Figure 1b. It would more useful to label the blot with what was blotted with. There is no SMART antibody so it'd be useful to identify how SMART expression was being detected. It doesn't appear to be MLKL because there's no endogenous band in WT samples.

RESPONSE: We used anti-GFP antibody that reacted with YFP portion of the SMART to detect the expression of SMART in various tissues. Thus, we have included the name of the antibody to detect the expression of SMART by Western blotting in the revised Figure 1b.

7. In figure 5e there are multiple comparisons between conditions for which significance is shown. It is not clear what the significant samples are being compared to. Bars should be put above the plots to show what 2 samples are being compared and whether the difference is significant.

RESPONSE: Sorry for the ambiguous presentation of the Figures. To clarify the samples being compared, we have added the bars to indicate which are compared in the revised Figure 5.

8. Some more details required in Methods. The source of MLKL ko MEFs and a citation to allow others to repeat the study as required; are the mice N or J background?; why 20mg/kg cisplatin? The data to find this as an optimal concentration are not shown and I feel would be useful for the readership; Line 439 – what anesthetic was used?

RESPONSE: As suggested, we have included the detailed description how we prepared MEFs from SMART Tg, wild-type, and *Mlkl*^{-/-} mice as follows: “MEFs were prepared from mice of the indicated genotypes at E13.5 - E14.5 after coitus using a standard method as described previously.” (Page 23, lines 368-372).

We generated SMART Tg mice on C57BL/6N genetic background (Page 24, line 386).

To induce acute kidney injury with cisplatin, as far as we investigated, the most popular doses of cisplatin were 20 mg/kg of body weight of mice (Hu et al., 2022; Meng et al., 2018; Wang et al., 2019; Xu et al., 2015; Zhang et al., 2007). We have also found only one paper that used cisplatin at 40 mg/kg of body weight of mice (Liu et al., 2006). Therefore, 20 mg/kg is relevant to investigate cisplatin-induced kidney injury in mice.

Regarding the anesthetic used in intravital mouse imaging, we used isoflurane via inhalation and have described it in the Methods as follows “Then, mice were anesthetized with isoflurane (1.5% inhalation, 0.5 L/min).” (Page 31, lines 519-520).

Minor

Some plurals are incorrect throughout, e.g. findings on line 160 and conditions on line 155; period on lines 223 and 231 rather than periods. I’m sure these will be easily corrected in proof/copyediting however. A typo of dilation on line 197.

RESPONSE: As suggested, we have corrected the typos as indicated (Page, 11, line 167 and Page 15, lines 244 and 251).

Line 263 – delete relatively as it is unclear what it is relative to

RESPONSE: As suggested, we have deleted “relatively” (Page 18, line 294).

Line 287 – it would be constructive for readers if the authors could specify how the ripoptosome differs from the necrosome which is more conventionally associated with necroptosis.

RESPONSE: As suggested, we have described the Ripoptosome in detail (Pages 19-20, lines 318-322)

Line 293. Please specify the RIP1 inhibitor. Not all operate via the same mode of action

RESPONSE: As suggested, we changed the sentence to “the RIPK1 inhibitor, Necrostatin-1s” (Page 20, line 326).

Line 296. RIPK3^{-/-} should have superscript ^{-/-}

RESPONSE: As suggested, we have changed *Ripk3^{-/-}* to *Ripk3^{-/-}* throughout the manuscript.

Reviewer #2 (Remarks to the Author):

In their paper Shin Murai et al., extended their model of measuring RIPK3 activity from in-vitro into in-vivo model by introducing their FRET biosensor, SMART into mice to generate a reporter mice. This is very important tool, which is very much needed, for the growing research on necroptosis. In general the data and methodologies during the manuscript are appropriate and are clearly presented.

There is one major issue with the data throughout the all manuscript, which is the claim that this is a specific system to monitor only necroptosis. Although, they have nicely showed that it can detect necroptosis and not other cell death mechanisms as apoptosis nor pyroptosis, they did not show that it is detecting necroptosis. In fact, their system detect the ability of phosphorylated RIPK3 (pRIPK3) to bind the binding domain of the necroptotic executor MLKL. This is very important as for example, if they will repeat their experiments using an MLKL KO mice or cells, they will probably detect necroptosis, which of course will be wrong, as necroptosis can't be executed without MLKL. Thus, in cells that don't express sufficient MLKL this system may results in false positive result, as it will measure RIPK3 activity.

Therefore, I will suggest that the authors will correct their text (including title and abstract) to clearly state that this system monitor RIPK3. They should discuss that as necroptosis is known, as for today, to be the main downstream pathway of pRIPK3 oligomerization, there system is can be used to analyze necroptosis, but it should be confirmed with other methods as specifically looking on MLKL phosphorylation or using MLKL KO.

RESPONSE:

We apologize that we did not describe the mechanisms clearly enough. We have now elaborated more on the mechanisms underlying the FRET response of SMART. We have also toned down some descriptions, such as “SMART senses necroptosis”, because we have only showed that SMART sensed conformational changes of MLKL, but not necroptosis itself.

Regarding the molecules sensed by SMART, we previously reported that oligomers of MLKL increase the FRET/CFP ratio in several cell lines (Murai et al., 2018). However, in the initially-submitted manuscript, we only analyzed SMART *Ripk3*^{-/-} mice, but not SMART *Mlkl*^{-/-} mice. Therefore, we only stressed that phosphorylation of RIPK3 was important for the increase in the FRET/CFP ratio in primary macrophages and renal tubular epithelial cells. To substantiate our previous model (Murai et al., 2018), we generated SMART *Mlkl*^{-/-} mice and analyzed peritoneal macrophages and renal tubular epithelial cells. The increase in the FRET/CFP ratio was abolished in BV6/zVAD-treated

macrophages from SMART We apologize that we did not describe the mechanisms clearly enough. Regarding the molecules sensed by SMART, we previously reported that oligomers of MLKL increase the FRET/CFP ratio of SMART in several cell lines (Murai et al., 2018). However, in the initially-submitted manuscript, we only analyzed SMART *Ripk3*^{-/-} mice, but not SMART *Mlkl*^{-/-} mice. Therefore, we only stressed that phosphorylation of RIPK3 was important for the increase in the FRET/CFP ratio in primary macrophages and renal tubular epithelial cells. To substantiate our previous model (Murai et al., 2018), we generated SMART *Mlkl*^{-/-} mice and analyzed peritoneal macrophages and renal tubular epithelial cells. The increase in the FRET/CFP ratio was abolished in BV6/zVAD-treated macrophages from SMART *Mlkl*^{-/-} mice (revised Figure 3a, b, c). Surprisingly, the averaged FRET/CFP ratios were higher in renal tubular epithelial cells of cisplatin-treated SMART *Mlkl*^{-/-} mice than those of untreated SMART *Mlkl*^{-/-} mice (revised Figure 6h, i). Related to this observation, we found that overexpressed SMART interacted with MLKL and RIPK3 in HEK293 cells (new Supplementary Figure 3). These results suggest that SMART may have the ability, at least in part, to sense conformational changes of RIPK3 through the interaction with RIPK3 under *Mlkl*-deficient cells. However, considering our previous results (Murai et al., 2018) and the present results (revised Figures 1 to 3), we assume that SMART mainly senses conformational changes of MLKL by interacting with MLKL under *Mlkl*-sufficient cells. Accordingly, we have revised Figure 1a to include the current our model and Figures 3 and 6h, i to include new results using SMART *Mlkl*^{-/-} mice. In addition, we have made new Supplementary Figure 3 to support the results using cisplatin-treated SMART *Mlkl*^{-/-} mice. We have mentioned these in the revised manuscript (Page 3, lines 42, 44, Pages 6 to 7, lines 100-109, Page 8, lines 116-119, Pages 11 to 12, lines 171-184, Page 16, lines 256-264, Pages 17-18, lines 276-290, Page 41, lines 777-782).

As suggested by the reviewer, we have changed the title to “Generation of transgenic mice expressing a FRET biosensor, SMART, that responds to necroptosis”. (Page 1, lines 1-2).

Reviewer #3 (Remarks to the Author):

In the presented manuscript, Murai et al. present their work on a FRET biosensor to allow necroptosis assessment in kidney tubular cells and in a mouse model of cisplatin-induced AKI. They refer to their technology as sensor for MLKL activation based on FRET “SMART” – this was published in NatComm in 2018 - and now generated SMART transgenic mice. They now investigated these mice with what they refer to as live cell imaging for secretion (LCI-S) of DAMPs. That latter system is based on a sandwich ELISA and allows them to detect HMGB1-release in parallel to necroptosis monitoring. This technology will be very helpful to many laboratories. More importantly, however, this manuscript refers adds important evidence to the role of necroptosis in acute kidney injury, and in particular in tubular cells. Authors should point out that so far, necroptosis was detected in human biopsy samples from AKI patients only by immunohistochemistry (Gong et al., Cell, 2017, PMID 28388412) which is hard to control in humans. Adding this novel assay convincingly supports the initial notion of necroptosis in tubular cells. Not only that, maybe this technology exhibits the most convincing piece of evidence for necroptosis involvement in AKI. That said, I have a couple of very minor concerns regarding this manuscript that in my understanding must not be held back from the scientific community for any reason. It should be published as soon as possible, and the tool should be made available to scientists working in this field.

RESPONSE: Thank you very much for the positive comments and for encouraging us to work on our project using SMART Tg mice.

Minor concerns

1) Reference #36 is problematic as the same group has toned down their conclusion of this paper in a subsequent manuscript. Especially when citing this manuscript for the effects observed by RIPK kinase inhibitors, I recommend caution.

RESPONSE: Thank you for the thoughtful advice. Indeed, we found that the numbers of pRIPK3⁺ cells were not dramatically increased in the intestine following TNF injection. At this moment, we have not obtained consistent results

of the FRET response of SMART using a TNF-induced SIRS model with two-photon excitation microscopy due to the following reasons.

- 1) First, as mentioned in the revised manuscript, only small numbers of intestinal epithelial cells expressed phospho-RIPK3 (a hallmark of necroptosis) in wild-type mice following high doses of TNF injection (0.5 mg/kg) (Page 13, lines 208-209 and Supplementary Figure. 2a).
- 2) Second, it was difficult to measure continuously the FRET/CFP ratio of SMART in an individual epithelial cell with two-photon excitation microscopy due to constant peristalsis of the small intestine (Supplementary Movie 4).
- 3) Third, we usually ligated 2 cm long ileum at the rostral and caudal sides and filled the ligated ileum with PBS to reduce peristalsis. This ligation gradually induced ischemia, resulting in an increased FRET/CFP ratio of many epithelial cells at later time points after TNF injection (Supplementary Figure. 2b, c, and Supplementary Movie 4).

To include the results of a TNF-induced SIRS model, we have made Supplementary Figure 2 and Supplementary movie 4 and mentioned these in the revised manuscript (Page 13, lines 204-212).

2) The occurrence of secondary apoptotic cells in Fig. 5d might be best interpreted as a regenerative reaction that only comes in at day 3 following cisplatin. Inhibition of apoptosis in this model does not result in protection from the increase in BUN or serum creatinine concentrations.

RESPONSE: Thank you for the insightful comments.

3) The intravital imaging is fantastic, congratulations!!!

RESPONSE: Thank you for your appreciation of our work.

Rebuttal References

- Hu, J., Gu, W., Ma, N., Fan, X., and Ci, X. (2022). Leonurine alleviates ferroptosis in cisplatin-induced acute kidney injury by activating the Nrf2 signalling pathway. *Br J Pharmacol*.
- Liu, M., Chien, C.C., Burne-Taney, M., Molls, R.R., Racusen, L.C., Colvin, R.B., and Rabb, H. (2006). A pathophysiologic role for T lymphocytes in murine acute cisplatin nephrotoxicity. *J Am Soc*

Nephrol 17, 765-774.

Meng, X.M., Li, H.D., Wu, W.F., Ming-Kuen Tang, P., Ren, G.L., Gao, L., Li, X.F., Yang, Y., Xu, T., Ma, T.T., et al. (2018). Wogonin protects against cisplatin-induced acute kidney injury by targeting RIPK1-mediated necroptosis. *Lab Invest* 98, 79-94.

Murai, S., Yamaguchi, Y., Shirasaki, Y., Yamagishi, M., Shindo, R., Hildebrand, J.M., Miura, R., Nakabayashi, O., Totsuka, M., Tomida, T., et al. (2018). A FRET biosensor for necroptosis uncovers two different modes of the release of DAMPs. *Nat Commun* 9, 4457.

Wang, J.N., Liu, M.M., Wang, F., Wei, B., Yang, Q., Cai, Y.T., Chen, X., Liu, X.Q., Jiang, L., Li, C., et al. (2019). RIPK1 inhibitor Cpd-71 attenuates renal dysfunction in cisplatin-treated mice via attenuating necroptosis, inflammation and oxidative stress. *Clin Sci (Lond)* 133, 1609-1627.

Xu, Y., Ma, H., Shao, J., Wu, J., Zhou, L., Zhang, Z., Wang, Y., Huang, Z., Ren, J., Liu, S., et al. (2015). A Role for Tubular Necroptosis in Cisplatin-Induced AKI. *J Am Soc Nephrol* 26, 2647-2658.

Zhang, B., Ramesh, G., Norbury, C.C., and Reeves, W.B. (2007). Cisplatin-induced nephrotoxicity is mediated by tumor necrosis factor-alpha produced by renal parenchymal cells. *Kidney Int* 72, 37-44.

REVIEWERS' COMMENTS:

Reviewer #1 (Remarks to the Author):

The authors have done well to address all of my comments on those of the other reviewers. I would suggest some very minor revisions nonetheless. The Kinase-like domain of MLKL is better known in the field as a pseudokinase domain (as per the definition of Manning et al., 2002, and many who have followed). I realise this is somewhat of a personal choice of course, but for consistency with the literature, definition as a pseudokinase domain (even if referred to as a KLD elsewhere) is important.

In the revision, methods for the SIRS experiments were included. The mode of delivery – tail vein injection? – vehicle, and method of sacrifice were not stated. I feel these are important inclusions if others were to replicate.

Reviewer #2 (Remarks to the Author):

In their revised manuscript Shin Murai et al. performed all my required experiments, mainly generating new SMART mice by crossing them into the MLKL KO and reacting their major experiment models.

Thus, their new results, support their hypothesis with a small exception regarding the specificity of the system to only sense necroptosis, as it seems that in the MLKL KOs kidney experiments they had a response, suggesting that is some tissue or cell that will not have MLKL, you can get a false positive response for necropolis readout.

Nevertheless, with the correct changes in the text for this small limitation of the use of mice, this is a very important study and model for the field.

I will only ask for the authors to also include this limitation in their abstract.

Reviewer #3 (Remarks to the Author):

The authors have further improved their manuscript. I consider this appropriate for immediate publication and congratulate the authors to generating an important new tool.

REVIEWERS' COMMENTS:

Reviewer #1 (Remarks to the Author):

The authors have done well to address all of my comments on those of the other reviewers. I would suggest some very minor revisions nonetheless. The Kinase-like domain of MLKL is better known in the field as a pseudokinase domain (as per the definition of Manning et al., 2002, and many who have followed). I realise this is somewhat of a personal choice of course, but for consistency with the literature, definition as a pseudokinase domain (even if referred to as a KLD elsewhere) is important.

RESPONSE: Thank you for the thoughtful suggestion. As suggested, we have changed to “a pseudokinase domain”(Page 5, line 75; page 42, line 810).

In the revision, methods for the SIRS experiments were included. The mode of delivery – tail vein injection? – vehicle, and method of sacrifice were not stated. I feel these are important inclusions if others were to replicate.

RESPONSE: As suggested, we have changed the description to “We injected mice with mTNF (0.5 mg/kg in PBS) via the tail vein and sacrificed them by cervical dislocation.” (Page 29, line 494 to page 30, line 495; page 33, lines 556-557).

Reviewer #2 (Remarks to the Author):

In their revised manuscript Shin Murai et al. performed all my required experiments, mainly generating new SMART mice by crossing them into the MLKL KO and reacting their major experiment models. Thus, their new results, support their hypothesis with a small exception regarding the specificity of the system to only sense necroptosis, as it seems that in the MLKL KOs kidney experiments they had a response, suggesting that is some tissue or cell that will not have MLKL, you can get

**a false positive response for necropolis readout.
Nevertheless, with the correct changes in the text for this small limitation of the use of mice, this is a very important study and model for the field.**

I will only ask for the authors to also include this limitation in their abstract.

RESPONSE: Thank you for your suggestion of limitations of the SMART biosensor. Due to the word length of the abstract (up to 150 words), we could not include such a sentence in the abstract. To circumvent this issue, we have mentioned limitations of the study at the end of the Discussion (Page 21, lines 349-353).

Reviewer #3 (Remarks to the Author):

The authors have further improved their manuscript. I consider this appropriate for immediate publication and congratulate the authors to generating an important new tool.

RESPONSE: Thank you for your appreciation of our work.